



# Assimilation of passive microwave vegetation optical depth in LDAS-Monde: a case study over the continental US

Anthony Mucia[1], Bertrand Bonan[1], Clément Albergel[1, 2], Yongjun Zheng[1], and Jean-Christophe Calvet[1]

[1]CNRM, Université de Toulouse, Météo-France, CNRS, 31057, Toulouse, France
[2]European Space Agency Climate Office, ECSAT, Harwell Campus, Oxforshire, Didcot OX11 0FD, UK

**Correspondence:** Jean-Christophe Calvet (jean-christophe.calvet@meteo.fr)

**Abstract.** The land data assimilation system, LDAS-Monde, developed by the Research Department of the French Meteorological service (Centre National de Recherches Météorologiques - CNRM) is capable of well representing Land Surface Variables (LSVs) from regional to global scales. It jointly assimilates satellite-derived observations of leaf area index (LAI) and surface soil moisture (SSM) into the Interactions between Soil Biosphere and Atmosphere (ISBA) land surface model (LSM),
increasing the accuracy of the model simulations and forecasts of the LSVs. The assimilation of vegetation variables directly impacts RZSM through seven control variables consisting in soil moisture of seven soil layers from the soil surface to 1 m depth. This capability is particularly useful in dry conditions, where SSM and RZSM are decoupled to a large extent. However, this positive impact does not reach its full potential due to the low temporal availability of optical-based LAI observations, at best, every ten days, and can suffer from months of no data over regions and seasons with heavy cloud cover such as winter
or monsoon conditions. In that context, this study investigates the assimilation of low frequency passive microwave vegetation optical depth (VOD), available in almost all weather conditions, as a proxy of LAI. The Vegetation Optical Depth Climate Archive (VODCA) dataset provides near-daily observations of vegetation conditions, far more frequently than optical based product such as LAI. This study's goal is to convert the more frequent X-band VOD observations into proxy-LAI observations through linear re-scaling and to assimilate them in place of direct LAI observations. Seven assimilation experiments were run
from 2003 to 2018 over the contiguous United States (CONUS), with 1) no assimilation, the assimilation of 2) SSM, 3) LAI, 4) re-scaled VODX, 5) re-scaled VODX only when LAI observations available, 6) LAI + SSM, and 7) re-scaled VODX + SSM. This study analyzes these assimilation experiments by comparing to satellite derived observations and in situ measurements and is focused on the variables of LAI, SSM, gross primary production (GPP), and evapotranspiration (ET). Each experiment is driven by atmospheric forcing reanalysis from the European Centre for Medium-Range Weather Forecasts (ECMWF) ERA5.
Results showed improved representation of GPP and ET by assimilating re-scaled VOD in place of LAI. Additionally, the joint assimilation of vegetation related variables (i.e. LAI or re-scaled VOD) and SSM demonstrates a small improvement in the representation of soil moisture over the assimilation of any dataset by itself.



## 1 Introduction

The coming decades are predicted to experience increases in extreme weather and climate events, primarily due to anthro-
pogenic warming (The Core Writing Team IPCC, 2015; Masson-Delmotte et al., 2021). Notably among these events are
droughts and heatwaves, which will lead to significant environmental, societal and economic damage. Droughts are partic-
ularly detrimental and costly extreme events (Bruce, 1994; Obasi, 1994; Cook et al., 2007). Human-induced changes to the
climate have increased the number and intensity of agricultural and ecological droughts, as well as increasing evapotranspi-
ration over land in some regions (Masson-Delmotte et al., 2021). The widespread and costly impact of these events makes it
critical to accurately monitor and predict land surface variables (LSVs) linking droughts and heatwaves to society (Di Napoli
et al., 2019). Improved knowledge of current LSV conditions, as well as potential forecasts and warnings of conditions in the
coming days or weeks gives stakeholders more useful information in order to prepare for and mitigate these extreme events.

To this end, Earth observations (EOs) and modelling of LSVs has proved to be of high importance. Variables such as leaf
area index (LAI), gross primary production (GPP), surface soil moisture (SSM), root zone soil moisture (RZSM), and evapo-
transpiration (ET) are specifically of interest to agricultural producers in drought prone areas. Satellite-derived observations of
these variables have near global coverage, but may suffer from spatial and temporal gaps and cannot observe all LSVs of inter-
est (such as RZSM). Observational errors and processing of the data also leads to these observations not perfectly representing
current LSV conditions. In contrast to satellite observations, land surface models (LSM) are able to simulate LSV conditions
at better temporal frequencies, and these models also have the potential for forecasting LSVs. However, LSMs can never be
perfect representations of the real world due to insufficient model physics, non-perfect initial conditions, and quality of the
atmospheric forcing.

In an effort to improve the monitoring and forecasting of LSVs, it is possible to combine EOs and LSMs via data assimilation
(DA) and land data assimilation systems (LDAS). The assimilation of EOs provide the LSM with more accurate and realistic
initial conditions, while also continuously correcting for known and unknown model biases. The end result is a spatially and
temporally continuous output, with improved representation of LSVs.

Numerous LDASs already exist, among them the Global Land Data Assimilation System (GLDAS) (Rodell et al., 2004),
North American Land Data Assimilation System (NLDAS) (Xia et al., 2012b, a), Coupled Land and Vegetation Data Assim-
ilation System (CLVDAS) (Sawada et al., 2015), and the Famine Early Warning Systems Network (FEWS NET) Land Data
Assimilation System (FLDAS) (McNally et al., 2017). At the meteorological research department of Météo-France, CNRM
(Centre National de Recherches Météorologiques), LDAS-Monde (Albergel et al., 2017) was developed as an offline LDAS
able to sequentially and simultaneously assimilate LAI and SSM into the ISBA (interactions between soil, biosphere, and at-
mosphere) LSM (Noilhan and Mahfouf, 1996; Calvet et al., 1998, 2004; Gibelin et al., 2006; Barbu et al., 2014). LDAS-Monde
has been studied using the ability to combine EOs and LSMs at a global and regional scale, used for monitoring and predicting
LSV conditions (Albergel et al., 2017, 2018b, 2019, 2020; Tall et al., 2019; Bonan et al., 2020; Mucia et al., 2020).

A historic focus of LDASs has been monitoring soil moisture (SM) through the assimilation of observational products
derived from active microwave scatterometers or passive microwave radiometers (De Lannoy et al., 2019). More recently,





some of this focus has shifted towards variables monitoring vegetation and vegetation dynamics or the joint assimilation of SM and vegetation related variables. LAI, for example, can be constrained in the LSMs indirectly, by the assimilation of LSVs such as brightness temperature (Vreugdenhil et al., 2016; Sawada et al., 2020) and radar backscatter (Lievens et al., 2017;

Shamambo et al., 2019). Direct assimilation of satellite LAI observations in LDASs is also possible, with significant advances in the reconstruction of carbon cycles (Fox et al., 2018), different assimilation approaches at global scales (Ling et al., 2019), and even the assimilation of vegetation optical depth (VOD) re-scaled to match LAI observations (Kumar et al., 2020).

The quality and frequency of the EOs used in assimilation are of utmost importance. While LDAS-Monde has the capability to assimilate both SSM and LAI observations, LAI assimilation provides a greater impact on vegetation conditions and RZSM

compared to the assimilation of SSM (Barbu et al., 2014). However, being based on optical remote sensing, LAI may suffer from poor temporal frequency as cloudy conditions completely prohibits retrievals. LDAS-Monde in a baseline configuration, assimilates LAI and SSM such as those from Copernicus Global Land Services (CGLS). CGLS LAI values are given every ten days, where they have been averaged over that period to account for cloudy conditions (Copernicus Global Land Operations, 2019). However, in some regions and seasons where persistent cloud cover exists for long periods, there can be months between

valid LAI retrievals.

Knowing how strong of a positive impact the assimilation of LAI has on the simulation, but also recognizing the weakness of assimilating these observations only every ten days, an alternative observational variable was sought that would take full advantage of the power of vegetation data assimilation. To that end, this study investigates the assimilation of linearly re-scaled vegetation optical depth (VOD) as a proxy for LAI. Kumar et al. (2020) has already shown that VOD assimilation as

an LAI proxy is possible, with the linear re-scaling and assimilation into the Noah-MP LSM. Being derived from microwave radiation observations, VOD is a nearly all-weather parameter, passing through cloud cover almost unaffected. This allows for more frequent retrievals of VOD, when compared to LAI. The VOD dataset used in this study is the Vegetation Optial Depth Climate Archive (VODCA) (Moesinger et al., 2020), which combines the VOD retrievals from numerous sensors into spatially homogeneous series, extending the periods of data between sensors. With this combination, VOD retrievals are, on average

over the CONUS domain, between every 1 and 2 days. This study seeks to use the link between LAI and VOD, to transform VOD into a proxy of LAI and assimilate as LAI in LDAS-Monde.

The goal of this study is to examine the impact of the assimilation of the more frequent VOD observations, as well as investigate future uses of VOD data in the LDAS-Monde system. Section 2 of this article details LDAS-Monde, the satellite derived data used in our data assimilation, the independent observations used for evaluation, as well as outlining the experiment setup

and assimilation scenarios. Section 3 provides comparisons of LAI and VOD data as well as the results of these experiments and the evaluation against our measuring datasets. Section 4 goes into discussion about the meaning of the results, and how they can be interpreted. And section 5 provides conclusion on the work, as well as highlighting future work in the same direction.



## 2 Methodology

### 2.1 LDAS-Monde

LDAS-Monde (Albergel et al., 2017, 2020) is a Land Data Assimilation System using the ISBA LSM and a Simplified Extended Kalman Filter (SEKF) to assimilate satellite derived observations of vegetation and soil moisture, within the SURFEX (Surface Externalisée V8.1) system (Masson et al., 2013). This global system is capable of well representing LSVs, and has more recently been able to produce forecasts of LSVs based on atmospheric forecasts as forcing. This study uses LDAS-Monde in a configuration with SURFEX V8.0 and the ISBA-A-gs LSM multi-layer soil scheme.

LDAS-Monde is capable of assimilating observations to directly update 8 control variables comprised of LAI and 7 soil moisture layers from 1cm to 100cm depths. Additional variables are indirectly modified by the assimilation through their biophysical feedbacks in the LSM. Because each observation directly updates LAI and soil moisture layers, even the assimilation of LAI alone allows for an analysis of the soil moisture at the root zone (1-100cm). Table 1 provides details about the LDAS-Monde parameters used in this study.

For the assimilation method, LDAS-Monde uses the SEKF as the default data assimilation scheme, but experiments have also used an Ensemble Kalman Filter (EnKF) and an Ensemble Square Root Filter (EnSRF) (Fairbairn et al., 2017; Bonan et al., 2020) schemes. This SEKF is described in further detail in Albergel et al. (2017); Bonan et al. (2020).

After assimilation, prognostic equations representing the physical processes of the LSM evolve the control vector to the end of the 24 hour assimilation window. Observations from the previous 24 hours are then assimilated which then form the 105 initial state of the next 24 hour period. Simplification is performed by using fixed estimates of background error variances and covariances instead of calculating them at the beginning of each cycle. This is the step that differentiates an EKF and SEKF. In LDAS-Monde, error values are fixed as 20% of observed LAI values and at a constant 0.05 $m^3/m^3$ for SSM. This complexity reduction aligns and continues with previous studies (Mahfouf et al., 2009; Albergel et al., 2010; Barbu et al., 2011; Fairbairn et al., 2017) which demonstrate the benefits of the simplification.

### 2.1.1 ISBA land surface model


For nature tiles as determined by land use databases, the ISBA LSM simulates heat, carbon, water, and other surface fluxes. Included within ISBA are several individual components simulating snow, hydrology, soil, and vegetation in the land surface system. The version of ISBA in this work uses the 12 layer snow parameterization scheme (Boone and Etchevers, 2001; Decharme et al., 2016), which better represent snow compaction, soil temperature, and surface albedo than previous snow 115 schemes.

This study focuses on the evolution of vegetation, specifically concerned with vegetation responses in drought events, requiring accurately simulated vegetation and soil moisture dynamics. To that end, the ISBA-A-gs (Calvet et al., 1998; Calvet, 2000; Calvet et al., 2004) version, which introduces the simulation of vegetation photosynthesis and stomatal conductance as well as allowing for the calculation of $CO_2$ fluxes from photorespiration is used to conduct this study. Additionally, the NIT 120 option is employed (Calvet and Soussana, 2001; Gibelin et al., 2006), which allows for the simulation of above non-woody





ground biomass, both leaf and structural, as well as transition the LAI variable from being prescribed to diagnostic based on the leaf biomass. ISBA also specifies minimum thresholds for LAI, as LAI that falls below this limit in the model is unable to accurately increase in the subsequent growing season. For evergreen forests, this threshold is 1 $m^2$ $m^{-2}$, and for all other types of vegetation the threshold is 0.3 $m^2$ $m^{-2}$.

The ISBA-Diffusion (Boone et al., 2000; Decharme et al., 2011) soil component of ISBA is also used, which has a 14 layer grid with depths reaching to 12 m, (0.01, 0.04, 0.1, 0.2, 0.4, 0.6, 0.8, 1.0, 1.5, 2.0, 3.0, 5.0, 8.0, and 12.0 m). A mixed form of the Richards equation is used to describe water fluxes in the entire root zone. This multi-layer scheme also provides overall improved surface flux and temperature predictions, compared to the simplified base soil component of the model, primarily due to better parameterization of latent heat from soil freezes.

**2.1.2    Land cover**

The configuration of ISBA for this study uses the ECOCLIMAP Second Generation (SG) (Calvet and Champeaux, 2020) database, the evolution of ECOCLIMAP-II (Faroux et al., 2013). ECOCLIMAP-SG uses 12 land surface patch classes comprised of nine plant types (evergreen broadleaf trees, needle leaf trees, deciduous broadleaf trees, herbaceous, tropical herbaceous, wetlands, C3 crops, C4 crops, and C4 irrigated crops). The remaining three patch classes are the non-vegetation surfaces

of rocks, bare soil, and permanent snow and ice. LDAS-Monde converts urban surfaces to bare rock for use in ISBA. A map of dominant land cover from ECOCLIMAP is shown in Fig. S1.

**2.2    Atmospheric forcing**

The ISBA LSM uses atmospheric reanalyses as forcings. The meteorological variables of air temperature, wind speed, air specific humidity, atmospheric pressure, shortwave and longwave downwelling radiation, and liquid and solid precipitation are

ingested into the model, and are the driving force of the LSVs. This model allows vegetation biomass and LAI to be discretely represented, and simulates exchanges in $CO_2$, energy, and water fluxes between the land surface and the atmosphere. Through recent updates, LDAS-Monde can now run in forecast mode (Albergel et al., 2019, 2020; Mucia et al., 2020), where ISBA can accept daily forecasts and produce individual outputs for each of the forecast time steps.

This study uses atmospheric reanalyses from ECMWF's ERA5 (Hersbach et al., 2018, 2020) to drive ISBA. This dataset

provides hourly data, globally over a 0.25° x 0.25° grid. The ERA5 reanalysis is itself a product of data assimilation, combining model data and observations around the world to create this consistent dataset from 1950-present. ERA5 assimilates atmospheric observations every 12 hours, which updates to a new, more accurate forecast. Its uncertainty is measured by sampling a 10 member ensemble every 3 hours, and the mean and spread of the ensemble is pre-computed and provided to users. While not a real-time product, preliminary ERA5 data are available with an approximate 5 day delay, with a higher quality

controlled release after 2-3 months.



## 2.3 Assimilated satellite observations

This study jointly and separately assimilates three sets of satellite derived observations. Each variable and the associated observations are described in this section.

### 2.3.1 Surface soil moisture

SSM observations are taken from the European Space Agency's (ESA) Climate Change Initiative (CCI), merging SSM observations microwave radiometers and scatterometers, with daily temporal coverage from 1978. The SSM data are provided in volumetric ($m^3$ $m^{-3}$) and is at 0.25°x0.25° spatial resolution. Snow cover, freezing ground temperature, and dense vegetation all greatly effect SSM retrievals, and this dataset provides quality flags for those conditions. Importantly, high elevation is known to negatively affect retrieval quality, and thus pixels with an average altitude above 1500m above sea level are filtered

out. This elevation filter does eliminate a large portion of the western and central western United States, mostly from the Rocky Mountains. As with previous studies using ESA CCI SSM (Albergel et al., 2017), the SSM product has been transformed into a model equivalent SSM using a linear re-scaling approach in order to address potentially incorrect parameters of wilting point and field capacity.

### 2.3.2 Leaf area index

Leaf area index, or LAI, is the sum of the one-sided area of a leaf's surface per unit area of land (Watson, 1947). This index is a very useful metric, allowing for the comparison of vegetation types despite potentially different plant spacing. LAI has proven to be a key variable when dealing with plant physiology, especially at the canopy level (Breda, 2003), as well as being strongly linked to vegetation biomass (Friedl et al., 1994; Gitelson et al., 2003).

Assimilated LAI observations in this study come from the CGLS LAI V2 product (Copernicus Global Land Operations,

2019). The observations come from the SPOT/VGT and PROBA-V sensors. The top-of-canopy (TOC) reflectance is input into a neural network for instantaneous LAI estimates. The V2 algorithm then applies filtering, smoothing, gap filling, and temporal compositional techniques to derive consistent LAI estimates every 10 days (Verger et al., 2014). The product is also compared with various datasets following the CEOS Land Product Validation Group's guidelines to ensure consistency with other LAI datasets. CGLS LAI V2 is available at 1km x 1km spatial resolution and from 1999 to present.

### 2.3.3 Microwave vegetation optical depth


Previous implementations of LDAS-Monde have assimilated LAI that has been of direct estimations from optical observations. In order to test how to improve initial conditions of the model, vegetation optical depth is used and transformed into an LAI-proxy. This study applies the same re-scaling methodology as Kumar et al. (2020) on the VODCA VOD dataset, and adding the capabilities of directly updating the RZSM control variables, and the potential joint assimilation with SSM.

VOD itself is the measure of attenuation of microwave radiation passing through a vegetation canopy (Jackson and Schmugge, 1991). The attenuation, which is a function of microwave frequency, can also be directly linked to vegetation water content





(Jackson et al., 1982; Wigneron et al., 1993; Owe et al., 2001). Because VOD is a large wavelength microwave product, observations of it are nearly all-weather, able to pass through cloud cover almost unaffected. This allows for far more frequent VOD observations compared to optical LAI observations, which is demonstrated in Figure 1. For the same 2003-2018 period, LAI

observations from CGLS are outnumbered by VOD observations from VODCA by approximately a factor of 6.

VOD is comprised of attenuation from several components, primarily standing vegetation (which itself is composed of green leaf biomass, green structural biomass such as stems, and woody biomass such as trunks and branches), necromass (primarily litter), and water interception from rain or dew. Recently, VOD has been more closely examined in regards to interacting effects of vegetation dynamics. A deeper look into L-band VOD by Konings et al. (2016) revealed that it is proportional to total

vegetation water content and tweaks their retrieval algorithm to more accurately account for vegetation effects on soil moisture observations.

Figure 2 shows the time series response of CGLS LAI (green, solid), VODCA VODX (red, dashed), and VODCA VODC (blue, dotted) near Lincoln, Nebraska from 2003-2018. This pixel is composed primarily of C3 and C4 crops. LAI observations have a far more predictable and seasonal pattern. X-band VOD also is a stronger signal compared to C-band. The peaks are

relatively close in timing in this case, but can also be offset due to the difference in peak vegetation water content. While this figure demonstrates that there is a correlation between LAI and VOD, it also shows that one cannot be substituted for the other.

As was done in Kumar et al. (2020), VOD observations are linearly re-scaled to match observed LAI over the same period, in this case from the CGLS LAI dataset. A 3-month linear re-rescaling window was selected to best match the VODCA VOD to CGLS LAI over seasonal timescales. Re-scaling is required because the ISBA LSM cannot simulate VOD directly, and thus

we cannot assimilate VOD data directly into the model. As shown in the Figure 2 time series, as well as what was demonstrated in Albergel et al. (2018a), LAI and VOD observations are correlated and this relationship enables us to match the VOD to LAI observations and use the resulting product to assimilate in place of LAI in the model. As a final step, a 90-day rolling average is applied after the re-scaling to smooth the results, and allow for better performance of the assimilated data.

This study uses the X-band of the newly created Vegetation Optical Depth Climate Archive (VODCA) (Moesinger et al.,

2020). VODCA is a synthesis of various satellite sensors since 1987, and uses the Land Parameter Retrieval Model (LPRM) V6, which simultaneously retrieves and calculates soil moisture and VOD from horizontally and vertically polarized microwave observations (Mo et al., 1982; Meesters et al., 2005; Owe et al., 2008; van der Schalie et al., 2017). The dataset is comprised from the AMSR-E, AMSR-2, SSM/I, TMI, and WindSat sensors, and separates the syntheses into C, X, and Ku-Band VOD retrievals. Ku-band VOD from VODCA did not encompass the entire period of interest in this study, stopping in 2017. While

both the C and X-band VOD may suitably represent vegetation over a wider array of land cover, X-band VOD was ultimately chosen to be assimilated in this study as Kumar et al. (2020) found large improvements in ET estimations from X-band VOD assimilation relative to C-band VOD. Because the TMI sensor aboard the Tropical Rainfall Measuring Mission (TRMM) satellite is in a 35° inclination orbit, and thus does not encompass the entirety of the CONUS domain, and because in the X-Band of VODCA, TMI is the only sensor between 1998 and late 2002, the year 2003 was chosen as a starting point in order

to have full domain observations.





Each sensor source in VODCA is first processed by removing locations known to be influenced by radio frequency interference (RFI), removing observations where land surface temperature (LST) is below freezing (as due to the changing dielectric permittivity of water and ice the VOD cannot be accurately retrieved in frozen conditions), and removing negative values of VOD, which are data artifacts and not physically possible. Daytime retrievals were found to have higher errors than their night-

time counterparts, and thus only nighttime retrievals are used in VODCA. The sensor datasets are then individually matched based on the VOD band by using an improved cumulative distribution function matching scheme to correct for systematic differences between the sensors (details of the improvements found in Moesinger et al. (2020)). Finally, where multiple sensor observations are available, the bands are then merged via arithmetic mean. This VODCA dataset is available globally at 0.25° x 0.25° spatial resolution. Due to the different number of sensors depending on each VOD band (and geographic location), and

the timing of the satellite overpasses, the merged product provides observations for at least 40% of all days with at least one sensor, and upwards of 70% with two or more.

### 2.4   Independent evaluation observations

In addition to comparing the results to the assimilated data themselves, this study uses independent satellite derived sources of evapotranspiration (ET), gross primary production (GPP), and in situ observations of SSM.

### 2.4.1   ALEXI evapotranspiration

Evapotranspiration (ET) is a broad term including many individual components and sources of evaporation and transpiration. These components include leaf transpiration, bare-soil evaporation, interception loss, surface water evaporation, and sublimation. ET is also strongly coupled with ecosystem production (Law et al., 2002), which in turn is driven by water availability (Noy-Meir, 1973). Therefore, measuring and predicting ET can be a valuable asset in terms of monitoring and predicting

agricultural droughts.

The Atmosphere-Land Exchange Inverse (ALEXI) is a surface energy balance model, which calculates evapotranspiration (ET) from a two-source land surface representation of the energy budget (Anderson et al., 1997, 2007a, b, 2011). The land surface is treated as a combination of soil and vegetation in the model, with each having unique temperatures, fluxes, and coupling with the atmosphere. Thermal infrared bands from the Geostationary Operational Environmental Satellite (GOES)

sensors estimate land surface temperature (LST) and provide the driving force for ALEXI over the United States, with Meteosat Second Generation (MSG) providing data over Europe and Africa. Global products use the Geoland2 land cover database (Lacaze et al., 2010) to estimate LST. Regional vegetation cover is estimated from MODIS-derived LAI products. Aerodynamic and atmospheric boundary layer conditions are derived from North American Regional Reanalysis (NARR) (Mesinger et al., 2006), Weather Research and Forecasting model (WRF) (Skamarock et al., 2005), and the Modern-era Retrospective Analysis

for Research and Applications (MERRA) (Gelaro et al., 2017) for the US, European/African, and Global domains respectively. Finally, the University of Maryland's global landcover classification (Hansen et al., 2000) is used to define surface characteristics over all domains. The ALEXI ET product is available at a spatial resolution of 0.05° x 0.05° globally, and at 0.04° x 0.04° over CONUS.





### 2.4.2 FLUXCOM gross primary production

Gross primary production (GPP) is a measure of $CO_2$ assimilated into vegetation by photosynthesis. This sequestration of carbon plays an important role in the global carbon budget. GPP is indicative of vegetation conditions and photosynthetic activity, and is highly coupled to water, light, and soil nutrient availability. However, direct, global measures of GPP are not currently possible (Anav et al., 2015) and instead must be estimated by measurements of carbon exchange between the land surface and the atmosphere.

The global FLUXNET network is a vast organization of eddy covariance towers used to measure trace gas fluxes between the biosphere and atmosphere (Jung et al., 2009; Pastorello et al., 2020). Machine learning algorithms are then applied to the energy and gas fluxes, as well as meteorological variables, to estimate fluxes in GPP and Terrestrial Ecosystem Respiration (TER) (Reichstein et al., 2005; Baldocchi, 2008; LASSLOP et al., 2010). This network of in situ measurements are then taken and combined with MODIS imagery for quality control and feature selection, then put through several machine learning

approaches, and finally combining with seasonal gridded satellite and meteorological observations to generate global carbon and energy flux products, FLUXCOM (Tramontana et al., 2016; Jung et al., 2018). This study uses the global GPP product from FLUXCOM to evaluate the performance of vegetation parameters independent from the LAI assimilated by LDAS-Monde. FLUXCOM GPP is available globally at 0.5° x 0.5° resolution, and from 1980 to present, however this study only uses data up to 2013 due to lack of data access.

### 2.4.3 United States Climate Reference Network

The United States Climate Reference Network (USCRN) is a sustained network of climate monitoring stations maintained by the National Oceanic and Atmospheric Administration (NOAA) National Centers for Environmental Information (NCEI) (Diamond et al., 2013; Bell et al., 2013). The network contains 114 stations in the contiguous U.S. and provides high quality, long-term temperature, precipitation, solar radiation, wind speed, humidity, soil moisture, and soil temperature observations.

This study uses the soil temperature and soil moisture observations, which are provided sub-hourly.

At each site, USCRN places three plots of probe units at five different depths, 5, 10, 20, 50, and 100cm. The soil moisture probe measures the dielectric permittivity of the soil by observing reflected EM waves at 50MHz, which is then converted to volumetic soil moisture ($m^3$ $m^{-3}$) via a calibration equation. Sensor calibration is also performed annually. A thermistor is also placed alongside the soil moisture sensor at all plots and depths. An average at each depth is calculated from the three plots

every 5 minutes and output data are typically publicly available within an hour of the reading. Figure 3 shows the locations of the USCRN in situ observations.

Four sensor depths are selected and are matched to ISBA soil layers, 5cm (WG3), 20cm (WG_20), 50cm (WG6), and 100cm (WG8). This comparison uses the ISBA soil layers to directly compare against the point measurements of the USCRN, but it is important to keep in mind that WG3, WG_20, WG6, and WG8 are layers of soil. WG3 is from 5cm to 10cm, WG_20 is a

weighted average of WG4 and WG5 (as performed in Mucia et al. (2020)) representing 10cm to 40cm, WG6 is a layer from 40cm to 60cm, and WG8 is the 80cm to 100cm layer.



This study compares USCRN data to LDAS-Monde soil moisture between the years of 2011 and 2018. While the network was operational as early as 2005, 2011 was selected as the start of the comparison in order to maximize the number of stations, and homogenize the results of comparisons between stations. The station observations are processed and filtered, with the most

notable filters including the removal of any data with corresponding soil temperature observations at or below 4°C. Stations with less than 100 days of observations are also removed, as the scores proved too variable. Finally, only correlation scores with associated p-values less than or equal to 0.05 are retained.

## 2.5 Experimental Setup and Assessment

The experiments performed and reported in this study occur over the Contiguous United States (CONUS) from 2003 to 2018.

This domain, as shown in Figure 3, is defined by 20° to 50° North and 130° to 60° West. The year 2003 was chosen as the start date, as this is approximately when the TRMM mission (containing the TMI sensor used in VODCA) ceases to be the only functioning VOD dataset included in VODCA. Because of the limited geographic extent of this mission (only up to  35°N), the analysis would be skewed. Table 2 provides the experiment names used throughout to reference the assimilation setups, and briefly describes what data are assimilated for each one. Besides the OL, SEKF LAI, SEKF VODX, SEKF SSM, SEKF LAI

SSM, and SEKF VODX SSM, several experiments were run at the same time, but using modified observations of VODX. The SEKF VODX10 uses VODX observations from VODCA as before, but has filtered those observations to coincide only where and when LAI observations from CGLS exist. This is used to test whether the changes produced between SEKF LAI and SEKF VODX are truly from the more frequent assimilation, or from the quantifiable differences between matched VODX and LAI. If the SEKF VODX10 results are closely resembling SEKF LAI results, but SEKF VODX are far different, this indicates the

frequency of assimilated observations is the primary cause of those differences.

The primary statistical score used in this study is the Pearson's correlation coefficient ($R$). The average correlation as well as distribution of correlations can allow the quick assessment of improvement or degradation, and are consistent with previous studies of LSMs and LSVs. In addition to the correlation, a normalized information contribution (NIC) is calculated for R as shown in Equations 1. This NICR, following Kumar et al. (2009), is normalized and thus allow for inter-comparison while

accounting for differences between variables and regions.

When analyzing the statistical scores of the USCRN, several conditions are applied. First, frozen soil conditions are avoided by only calculating scores based from observations when temperature measurements are above 4°C. As ISBA separately calculates frozen and liquid soil moisture, when conditions are close freezing, there can be significant errors. Second, only stations with more than 100 observations (at the respective depths) are calculated for a sufficient number of data points. Finally,

p-values are calculated along side the correlations, and stations without p-values of significance, as defined by p-value > 0.05, are screened out.

Bootstrapping is also used to calculate confidence intervals and thus determine statistical significance between different experiments. Essentially, bootstrapping is the repeated removal of random points in the dataset, and recalculation of the desired score or variable. This study uses a constant 10,000 repeats to calculate the confidence intervals in order to generate a

sufficiently large number of samples. In this study, bootstrapping is applied to the statistics versus the USCRN network.





For several analyses, probability distribution functions (PDFs) are estimated from the distribution of correlation scores of individual gridcells. These PDFs are derived using a Gaussian kernel density estimation, with "Scott's Rule" calculating the appropriate smoothing bandwidth. These PDFs give a far smoother and readable estimation of correlations when compared to simple histograms. These PDFs are used in order to better visualize and compare the distribution of scores from several

experiments at once, specifically to highlight smaller differences between experiments, which may not be easily seen in histograms. In the supplement, Fig. S2a provides an example of a typical histogram along with b,the accompanying PDF for the distribution of LAI correlations for different experiments over CONUS, demonstrating that they are in fact representing the same distribution.

## 3 Results

### 3.1 Relationship between VOD and LAI

Before assimilating VOD observations, the X-band VOD (refered to as VODX from now on) data was compared against LAI observations, as well as LAI from the ISBA OL to determine their respective relationships. Additionally, VODX and LAI observations were analyzed over individual patch types, where more than 50% of the patch represents a single vegetation type. These analyses provide more information regarding the strength of the VODX-LAI relationship over different vegetation types.

Figure 4 presents a density scatter plots, representing all times and points when there is both VODX and LAI data over the growing seasons (April-September) of the 2003-2018 period. Linear regression and correlation scores have been plotted over the data. Logarithmic transformations to the VODX data were also applied (not shown), but with no significant increase in regression correlation or regression shape.

The LAI observations are moderately well correlated to VODX observations, as shown by the 0.66 R. The relationship of

VODX with LAI from ISBA (Fig. 4b) actually provides higher values, with an R of 0.8 . Visually, we can see at higher LAI values, there is a more dramatic increase in VODX, whereas in Fig. 4a, the VOD values seem almost as if they flatten out above 1 m$^2$ m$^{-2}$ LAI. This is partly seen in Fig 4b, but is also slightly compensated by progressively higher VOD, while eliminating low VOD values at high LAI. Additionally, Fig. 4b clearly shows certain artificial thresholds from ISBA, including the 0.3 m$^2$ m$^{-2}$ lower limit of LAI for most vegetation types and the 1 m$^2$ m$^{-2}$ lower limit for evergreen forests. These artifacts can be seen

at a wide range of VOD values, and are also visible in these graphics for the sub-domains. These lower limits in ISBA are often reached in winter and spring months when vegetation activity and LAI are low, with the included spring months of April and May likely being the cause of this effect in Fig. 4. This figure reinforces that there is in fact a positive relationship between VOD and LAI observations.

When this same relationship of VODX observations versus LAI observations is performed over dominant patch types (ac-

cording to ECOCLIMAP-SG), new insight is gained on vegetation types where the two variables are far more closely linked. Figure 5 displays a density scatter plot of the observations of VODX compared to LAI observations over six ECOCLIMAP-SG patch types, namely (a) deciduous forests, (b) coniferous forests, (c) C3 crops, (d) C4 crops, (e) C3 grasslands, and (f) irrigated crops. Areas with higher density (i.e. higher concentrations of observations of LAI and VOD) are in darker colors. Spatial





averages of when LAI and VOD observations are compared are also displayed with colored dots (approximately one dot per LAI observation) representing the season in which they were observed, with winter (cyan), spring (green), summer (red), and autumn (yellow).

Overall, for most vegetation types, there is a moderate, positive correlation between the two observations. The notable exceptions are coniferous forests and irrigated crops, producing negative or near-zero R's. The seasonality does play a strong part as well, and the figures show a clear separation of values according to seasons over the patches. Winter correlations are typically low for all vegetation types, and the cyan seen in the graphs are often clumped at low LAI values. Spring scores are on average increased, and contain a far wider range of LAI values, but similar range of VOD values. Summer and autumn see the highest correlation scores and are characterized by a wider range of LAI and VOD values. Notably, deciduous forests have a negative correlation during Summer months, but the autumn correlation is strongly positive.

The same analysis by patch is also performed using matched VODX in place of VODX observations. This matched VODX is the product of the linear re-scaling transformation of the observations into an LAI-proxy. As is demonstrated in Figure 6, the correlations are very strongly improved over the non-matched observations. Even so, a seasonal hysteresis pattern emerges over C3 and C4 crop patches, where autumn (yellow) matched VODX is visibly higher than at the same LAI values compared to other seasons. This transition from VODX to matched VODX does show that the transformation is viable and strongly improves the correlation. This matched VODX product is able to be assimilated as an LAI-proxy into LDAS-Monde.

## 3.2 Impact of assimilating matched VOD

### 3.2.1 Assessment with satellite derived observations

This section analyzes the impact of using and assimilating matched VODX as an LAI proxy in the LDAS-Monde system. For this analysis, the primary variables of interest are LAI, GPP, ET, and SSM. Out of the four, GPP, ET, and SSM are the truly independent datasets to compare to, when investigating the assimilation of LAI or VODX. Most main conclusions are then drawn from the analysis of these variables. Comparisons to LAI observations are still presented, but as the LAI was used in the assimilation itself, or to re-scale the VOD, it acts more as a benchmark for the assimilation.

The average monthly correlations are presented in Fig. 7 for the OL, SEKF LAI, SEKF VODX, and SEKF VODX10. First looking at LAI, Fig. 7a, the whole CONUS domain sees added value during the months of May through September when assimilating matched VODX in place of LAI. The rest of the year, the scores for SEKF VODX are slightly below that of SEKF LAI. The improvement in LAI correlation from assimilating VODX comes as a slight surprise, as this is comparing to the CGLS LAI observations that themselves were assimilated in SEKF LAI. Some potential explanations include the far more frequent assimilation of VOD during the summer months when LAI is most rapidly changing. The results of SEKF VODX10 also show definitively that it is the more frequent observations, and not the differences between LAI and matched VOD that are causing some improvement, as SEKF VODX10 is consistently lower than both SEKF LAI and SEKF VODX throughout the year. Additionally, this panel shows that any assimilation of VODX or LAI significantly improves correlations compared to the model by itself (OL).



For the variable of GPP, Fig. 7b, some similar trends are seen. During the months of March through July, the assimilation of VODX performs far better than SEKF LAI or SEKF VODX10. From July to October, there is also some improvement, but not as strongly as in the spring and early summer. Interestingly, for the OL, SEKF LAI, and SEKF VODX10, there is a visible
dip in correlation scores during the month of May, while the SEKF VODX and the interpolated counterpart see near constant, or even slightly higher correlations compared to previous and future months. On average, the month of May sees some of the fastest vegetation change of the year for CONUS, and all the model, and even all the observations assimilated at best every 10 days may not provide sufficient constraint to the vegetation. The near daily VODX products do provide that, and thus seem to prove immediately their utility in use as LAI proxies for data assimilation. The changes in correlations between experiments
and GPP observations are not as drastic as seen in LAI, but they do show the same overall trends. And importantly, this GPP observation is an independent evaluation of vegetation conditions, where the larges improvements from VOD assimilation are observed in the spring and summer months, when droughts and heatwaves are most likely to damage agricultural production.

The ET variable, Fig. 7c, is harder to view differences, as the correlation scores for all the experiments seen here are relatively close. The only easily distinguishable differences arise in the months of May to August. During these months, like with LAI
and GPP, SEKF VODX has the highest correlations. It is followed by SEKF LAI and SEKF VODX10, which are close to one another, then finally followed by the OL. In general, these correlation scores are lower than for LAI or GPP, but also are at their peak during much of the summer, when evaporative demand is highest, and when it is critical for agricultural production to account for hot and dry conditions.

With SSM, Fig. 7d, correlations between the experiments are nearly indistinguishable except for some summer months,
namely June through August. Overall correlations are very high, consistently higher than 0.80, and contrary to all the other variables, provide the best correlations in winter months. Where we do see differences between the experiments between June and August, it can be noted that the OL actually performs the best, followed by SEKF LAI and SEKF VODX10, and finally with the lowest scores given to SEKF VODX. While in absolute terms these differences are small, a logical explanation can support these rankings: any data assimilation in LDAS-Monde, whether it is of vegetation such as LAI or VOD, or SSM,
directly changes the 8 control variables. Seven of these variables are soil moisture, with 6 of them deeper than the 5cm WG3 layer used to compare against these ESA SSM observations. The assimilation, of LAI or VODX in this case, impacts all these layers and can adjust the uppermost layer used here to coincide with higher LAI values. In these experiments, only the vegetation variable is assimilated, and thus there is no secondary compensation at the upper soil levels done by assimilating SSM observations.

The PDFs (derived from the histogram of correlation scores) are given in Figure 8 over CONUS. LAI, Fig. 8a, provides a very clear indication that the assimilation of vegetation variables, whether LAI or VODX heavily shifts the distribution of correlations higher. From correlations of 0.1 to 0.45, SEKF VODX has slightly more points than SEKF LAI, and this is reversed from 0.45 to 0.75. The SEKF VODX then quickly spikes at a correlation value of 0.88, with SEKF LAI shifted higher, with a peak closer to 0.9. These strongly changed values from the OL and similarities between SEKF LAI and SEKF VODX
demonstrate even more that VODX can properly act as an LAI proxy.





The distribution of GPP correlations, Fig. 8b, are not as widespread as LAI, however, a clear pattern still emerges. Starting at 0.4, the SEKF LAI and SEKF VODX have less gridcells than the OL, which lasts until around 0.8. It is around this point that the shift towards more higher correlation points with SEKF LAI and SEKF VODX is strongly apparent. While very similar, SEKF VODX does slightly outperform SEKF LAI in this case as well, having more higher correlation values.

Like with the monthly correlations presented before, this distribution of ET scores, Fig. 8c, are very similar between all the experiments. At around 0.4, there is a noticeable difference where SEKF VODX and SEKF LAI begin containing less points, which is then made up with those experiments having more higher values consistently between 0.55 and 0.7. At their peak densities, SEKF LAI slightly outperforms SEKF VODX, but SEKF VODX still improves over the OL.

For all displayed experiments, the distribution of SSM correlation scores in Fig. 8 (d) are almost bimodal. There is one peak
at 0.55 and another between 0.7 and 0.8. The experiments of SEKF LAI and SEKF VODX are only slightly different, with both edging out better performance over the OL. SEKF VODX also edges out a better performance than SEKF LAI, again with a slightly larger number of high correlations.

### 3.2.2    Assessment with in situ observations

The comparison of LAI, GPP, ET, and even SSM against satellite derived observations serves an important purpose, as those
observations are spatially continuous. However, errors in the sensors or processing of the data still exist, and relatively large spatial resolutions mean losses of more localized information. With this in mind, all of the experiments listed in Table 2 are also compared to soil moisture observations from the United States Climate Reference Network (USCRN) in situ soil moisture monitoring stations.

Table 3 provides the average correlations to the in situ stations for each of the experiments and at each of the depths. As
previously seen in Mucia et al. (2020), correlations strongly drop as the depths become lower. With 2 significant figures, the correlations at 5cm are all identical at 0.75. There is very slightly more variability at lower depths with 20cm scores ranging from 0.68 for the OL, to 0.70 for all the experiments jointly assimilating vegetation and soil moisture observations. Similar variations are present for the 50 and 100cm depths. Notably, the 100cm scores are all the same at 0.48 except for the OL and SEKF SSM experiments with 0.46.

Statistical bootstrapping was performed on all the calculated values, before rounding significant figures, to calculate the upper and lower bounds of the 95% confidence intervals (CIs). The resulting CIs showed that at every depth, all the experiment's mean correlations were within every CI, meaning no experiment could be said to be statistically different than any other at the same depth.

Seen throughout these results is the changing number of stations ($n$) used in the analyses. The 5cm comparisons have
110 stations, 20cm uses 87 stations, 50cm uses 85 stations, and 100cm uses 84 stations. This is simply due to the fact that the USCRN network cannot install soil moisture or soil temperature probes in hard or rocky ground layers, as stated in the USCRN soil climate observations documentation (USCRN). In all cases, the 5cm and 10cm probes are installed, but deeper layers depend on the regolith type.



To better analyze differences on a more individual scale, the normalized information contribution of the correlations (NICR)
were calculated for each experiment and each depth in comparison to the OL. These NICR values tell use by how much the
assimilation experiments improved or degraded scores in respect to the OL. Table 4 displays the each experiment and depth,
and the number of stations that were degraded (red), neutral (black), and improved (green). This approach avoids averaging
scores, while still providing a performance overview of the whole domain.

In a similar manner, Fig. 9 displays the PDF of the distribution of differences in correlation for each of the four depths
compared to the OL, and looks at the responses of SEKF SSM, SEKF LAI, SEKF VODX, SEKF LAI SSM, and SEKF VODX
SSM experiments.

In regards to the impact of assimilating re-scaled VOD in lieu of LAI, the NICR scores to the in situ observations provide
evidence that SEKF VODX and SEKF LAI are similar and comparable. The SEKF VODX increases the number of improved
stations over SEKF LAI at all depths, while also keeping constant or slightly increasing the number of degraded stations.
The improved stations do outnumber the degraded at all depths. Additionally, the SEKF VODX10 experiment shows stronger
similarities to SEKF LAI, than to SEKF VODX, indicating that the differences are due to the more frequent assimilation of
VODX. This results demonstrate that re-scaled VODX can indeed be a suitable substitute for LAI in LDAS-Monde.

### 3.3   Impact of individual and joint assimilation of vegetation variables and SSM

While previously discussed above when assessing correlations and NICR for the USCRN, this section will go into more detail
regarding the effects of jointly assimilating variables of vegetation (LAI or Matched VODX) and SSM in LDAS-Monde. We
have already seen that the joint assimilation provides a noticeable increase in improved USCRN stations relative to the OL,
over the single assimilation of vegetation variables or SSM.

Figure 12 presents the same type of figure as previously, looking at the monthly scores of four LSVs of interest over CONUS,
but including the joint assimilation experiments of SEKF LAI SSM and SEKF VODX SSM. The main concern in this figure and
the following, is determining the improvement, if any, between the dashed (single assimilation) and solid (joint assimilation)
red and green lines. As seen in panel a, there is no discernible difference between the single and joint assimilation for LAI.
But as panels b and c show, jointly assimilating vegetation and SSM produces slightly improved monthly correlations over the
whole CONUS domain for GPP and ET respectively. These improvements are primarily seen in the months of June through
August, and while the changes are small, they consistent improvements over the single assimilation of LAI or matched VODX.
Regarding the variable of SSM, the joint assimilation strongly improves the monthly correlations from LAI to LAI SSM and
from VODX to VODX SSM. To reiterate, because the SSM observations assimilated are the same used to compare in panel d,
it is merely an indication that the data assimilation is truly shifting model soil moisture values closer to that of observations.

When assessing with in situ observations, both the NICR table and PDFs, the 5cm changes are the strongest seen, with SEKF
VODX SSM providing the highest number of improved stations, while having some of the lowest number of degraded stations.
SEKF LAI SSM and SEKF VODX10 SSM perform similarly, but with a reduction in improved stations. Assimilating just
LAI, VODX, VODX10, or VODX_Int consistently under-perform their matching joint assimilation experiments by increasing
the number of degraded stations, while having fewer improved stations. And finally, SEKF SSM, while showing the fewest





degraded stations, also has the fewest improved. As depths get lower, the numbers become closer. It is generally still seen that the joint assimilation of vegetation and soil moisture improve more stations than the individual assimilation, and the number

of stations degraded stays similar. These trends go to show that the joint assimilation has distinct added value in soil moisture monitoring, and will be discussed in more detail in the next section.

In order to assess any geographic patterns of the USCRN correlations, maps of the NICR of each station are plotted for the four depths looking at the SEKF SSM, SEKF LAI SSM, and SEKF VODX SSM experiments. Figure 10 displays WG3 maps in panels a through c and WG_20 maps in panels d through f. Figure 11 provides WG6 maps in panels a through c and WG8

maps in panels d through f. It is clear from these figures that stations are strongly improved by using either LAI SSM or VODX SSM assimilation, compared to just SSM. Additionally, a geographic pattern does emerge, primarily at 5 and 20cm. This pattern is that much of the Great Plains and Midwest show strong improvement due to the joint assimilation, while the south and east coasts show little change. Stations in the western US are more sporadic, with improved, degraded, or neutral stations spread throughout. This western region is, however, where most of the stations experiencing degradation are located. At 50

and 100cm, the SEKF SSM show minimal concrete changes compared to the OL, with near-even numbers of improved and degraded sites. The joint assimilation improves upon these values, and while the degraded stations are typically still degraded in SEKF LAI SSM and SEKF VODX SSM, more stations move from neutral to improved.

Overall, the use of the USCRN in situ observations agrees with the hypothesis that the assimilation of vegetation has stronger effects on soil moisture than the assimilation of SSM. It also goes to show that the assimilation of VODX is on par or even an

improvement from the assimilation of LAI.

## 4   Discussion

### 4.1   Does the assimilation of matched VOD improve LDAS-Monde's ability to monitor drought?

In the comparison of VOD and LAI before linear re-scaling, it is immediately apparent that vegetation type plays a large role in their relationship. These values seen and described in the results seem to indicate that heavily forested regions have only weak

correlations between VOD and LAI observations. Saatchi et al. (2011) demonstrates that L-band satellite radar estimations of above ground biomass (AGB) are strongly impacted by forest structure, and Mialon et al. (2020) shows poor correlations between L-band VOD and estimated AGB over heavily forested areas of the Northern hemisphere. This assessment, having isolated certain woodlands, lends credence to that same conclusion, that radar retrievals over forests have a more delicate and variable response.

After linear re-scaling the VOD to match the LAI observations, hysteresis patterns over C3 and C4 crops emerged. Due to these patches being dominated by cultivated vegetation, it is possible that heavier ground litter during and after harvesting produces a stronger VOD response while at lower LAI. This effect is more apparent in C4 crops potentially due to their overall higher LAI and VOD values. Leaf water interception, both on growing plants and ground litter, could also disproportionately increase the VOD observations, leading to an offset. Future studies could include filtering and analyzing this LAI-VOD

relationship accounting for leaf water interception. Additionally, VOD being more strongly related to leaf biomass than to



LAI could lead to a non-constant LAI to VOD ratio. This ratio could be related to specific leaf area (SLA), as pointed out by Shamambo et al. (2019). This effect would be most marked over near-uniform vegetation types, including some unmixed forests and many crops, and would emerge during the growing season. This matches the observed hysteresis most strongly seen in areas dominated by C3 and C4 crop types, and in the growing seasons of spring and summer, found in this study.

After assimilation, the SEKF VODX experiment provided either comparable or improved seasonal representation of GPP and ET, two important LSVs with regard to agricultural droughts. The distribution of correlations for these LSVs also agree that the assimilation of VODX provides approximately equal or improved correlations to GPP and ET over this domain. The assessment with the in situ USCRN, this idea is strengthened as SEKF VODX provided more improved stations at all soil depths. And even though the average correlations were statistically indistinguishable, the trend still showed some improvement
from SEKF VODX. Further studies comparing to in situ soil moisture with a stronger significance would be a suitable follow up to solidify this trend.

### 4.2    Does the joint assimilation of vegetation variables and soil moisture provide better initial conditions?

Over the entire CONUS domain, there is evidence indicating added value for GPP and ET variables to from joint vegetation and soil moisture assimilation. In conjunction with the results from the USCRN analysis, joint assimilation does overall show
more potential value than assimilation of vegetation related variables, and few, if any, drawbacks. Additionally, it is also shown that the assimilation of vegetation, be it LAI or matched VODX, has a far stronger impact on the LSVs of LAI, GPP, and ET compared to the assimilation of SSM.

Another step in the direction of improving initial conditions consists of directly assimilating level 1 data, such as ASCAT radar backscatter or microwave brightness temperature from passive instruments for example, into LDAS-Monde. This would
allow the use of all the information contained in the signal, including soil moisture and vegetation, and at the same time reduce observations errors in the data assimilation (Shamambo et al., 2019; Shamambo, 2020). A major step in this development is the solution of the observation operator, and current steps towards developing this operator are focused on using machine learning approaches to find the optimal term.

### 4.3    Towards future drought monitoring and forecasting with LDAS-Monde

With the assimilation of VOD, initial conditions of LDAS-Monde were shown to be generally improved. A next step is then to move towards using these improved initial conditions to better forecast drought impacts using LDAS-Monde in a forecast configuration. A framework of running LDAS-Monde forecast over a time and place with a known drought event can be envisioned.

Future studies with LDAS-Monde are proposed here, using the initializing the land surface forecasts via joint assimilation
of VOD and SSM. As shown in the results, this assimilation configuration provides the most frequent updating of land surface variables through data assimilation, and can more accurately track variables such as ET, RZSM, and GPP. LDAS-Monde variables such as these can be computed into a percentile rank to match other drought indices and monitors such as the U.S. Drought Monitor (Svoboda et al., 2002). This will be done by comparing the experiment with VOD and SSM assimilation to

that of an OL, run over a far longer time period. These percentiles can be directly compared to the USDM over a known drought

period, and statistical assessment (such as false alarm rate, probability of detection, and Spearman correlation) to determine

how accurate the land surface forecasts prove in an extreme event.

## 5    Conclusions

A generally positive, but still variable, relationship between LAI and VODX was found, where dominant vegetation is a strong

factor. This result agrees with previous work comparing vegetation cover and conditions to VOD. Coniferous forests consis-

tently had the weakest correlation, while C3 and C4 crops typically had the best. Seasonality also dramatically changed the

LAI-VODX relationship, with winter scores typically the lowest, and summer and autumn correlations typically the strongest.

Matched VODX was far more strongly correlated to LAI observations than non-matched VODX, as expected, but still exhibited

significant variation.

This study then demonstrated the utility of the assimilation of microwave X-band VOD into LDAS-Monde, as well as showed

a slight advantage by using a joint assimilation of vegetation related observations (VOD or LAI) in combination with SSM. This

assessment over the United States showed improved representation of ET and GPP, comparing to independent observations,

over some months assimilation of VOD. The improvements seen in GPP and ET correlations by assimilating matched VODX

in place of LAI are almost entirely due to the more frequent observations of VOD. This is shown because the experiment SEKF

VODX10, which assimilated matched VODX at the same frequency as LAI observations performs considerably worse than

the natural VODX observation frequency.

Using in situ observations of soil moisture, the joint assimilation of vegetation related observations and SSM reduced the

number of degraded correlations, while increasing the number of improved stations for the top 20cm of soil. Additionally,

when assessing the RZSM, the assimilation of SSM by itself proved to be weaker than that of vegetation (LAI or VOD) alone,

even at the uppermost layers of soil.

Follow up studies will bring these improvements into the LDAS-Monde forecast configuration, and will study the capability

of early detection and warning of drought events.

*Author contributions.*  AM, CA, and JCC conceptualized the project. AM led the investigation, determined the methodology, and wrote the

original draft of the paper. All co-authors contributed to the review and editing of the paper.

*Competing interests.*  The authors declare that they have no conflicts of interest.



*Acknowledgements.* This research has received funding from the French Make Our Planet Great Again program, and from the European Union Horizon 2020 research and innovation program under grant agreement No 958927 (CoCO2).



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





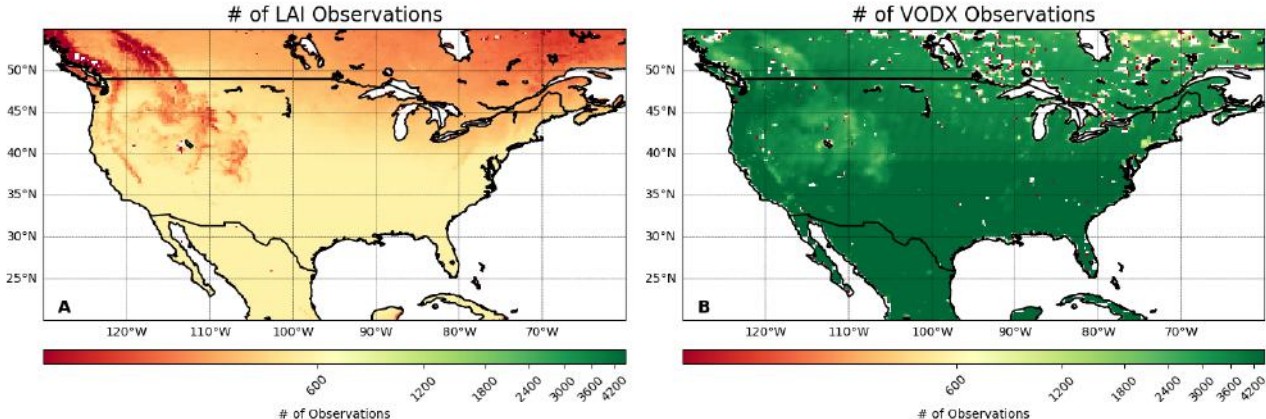

**Figure 1.** Maps showing the the cumulative number of observations provided for A) CGLS LAI and B) VODCA VODX over CONUS between 2003-2018. The colorbar is on a log scale in order to show the vast differences between the two datasets.





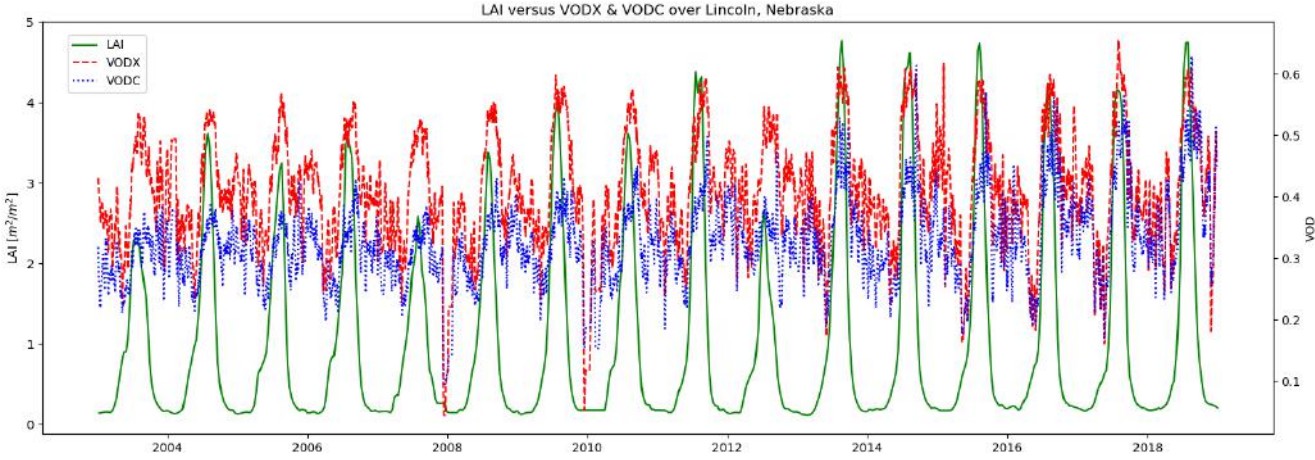

**Figure 2.** A time series showing the response of CGLS LAI (green, solid), VODCA VODX (red, dashed), and VODCA VODC (blue, dotted) over Lincoln Nebraska from 2003-2018.



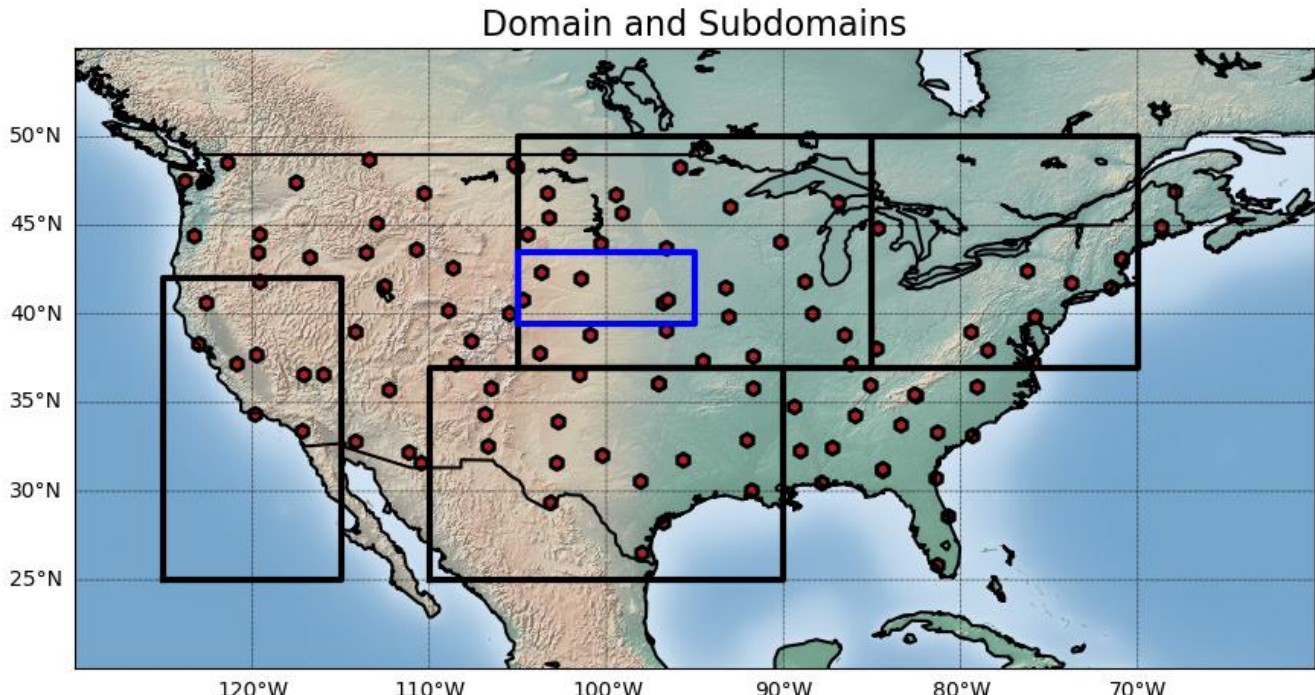

**Figure 3.** Map illustrating the CONUS domain, with boxes around selected sub-domains of interest. The blue box represents the Nebraska domain. Red dots represent the locations of the USCRN SM stations.



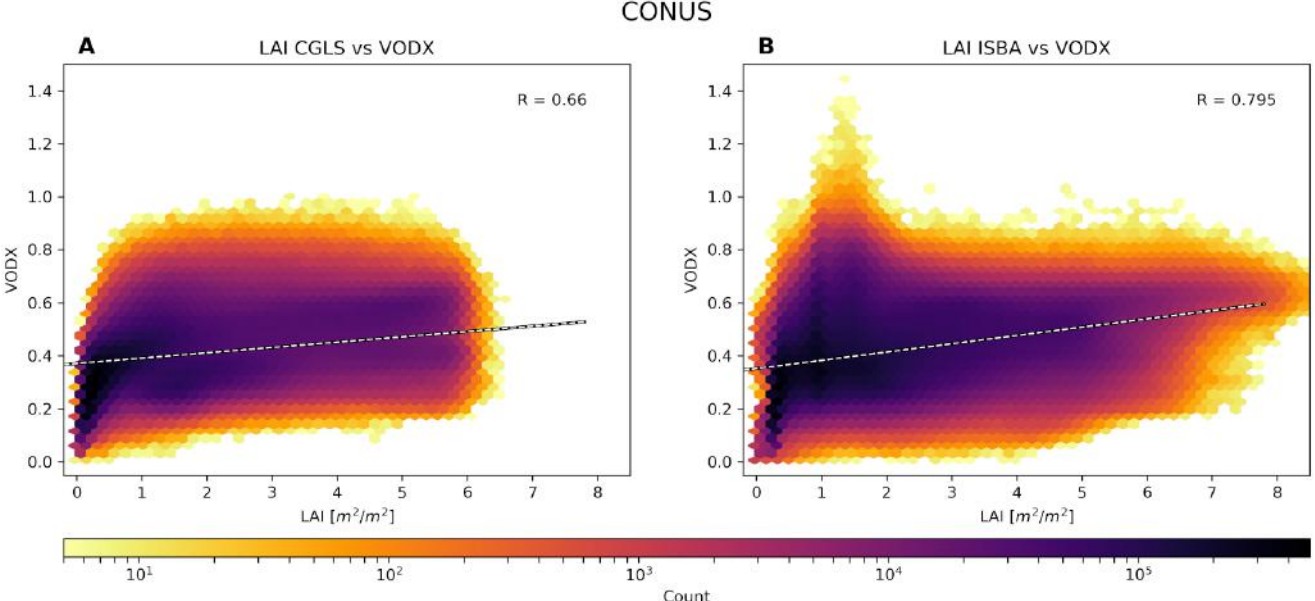

**Figure 4.** A density scatter plot detailing the relationship between LAI and VODX from VODCA. This comparison only analyzes where and when there are both LAI and VOD observations. Warm colors represent more counts of points in the hexagonal bins. The colorbar is logarithmically scaled in order to emphasize the distributions, and bins with under 5 count are eliminated. A regression line and correlation score are added. This comparison only looks at points during the growing season months (April-September). Panel A) compares LAI from CGLS observations while panel B) compares to LAI from ISBA OL.



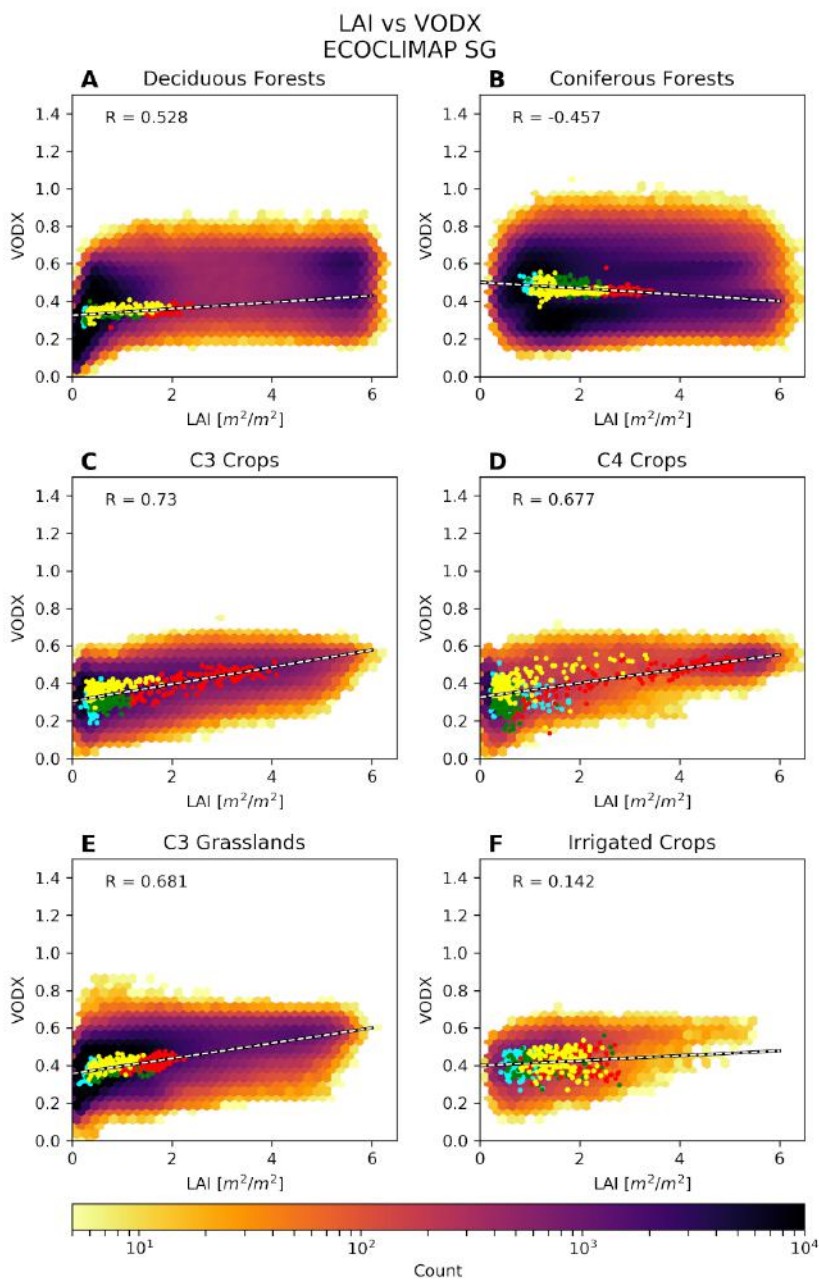

**Figure 5.** A density scatter plot detailing the relationship between LAI and VODX from VODCA over six dominant vegetation types, A) Deciduous Forests, B) Coniferous Forests, C) C3 Crops, D) C4 Crops, E) C3 Grasslands, and F) Irrigated Crops. Dominant vegetation is defined as where 50% or more of a patch containing a single vegetation type. Higher concentrations of points trend towards black. Colored dots represent the spatial average over the four seasons, where cyan is winter, green is spring, red is summer, and yellow is autumn. Black and white dashed lines represent the linear regression of the seasonal scores.

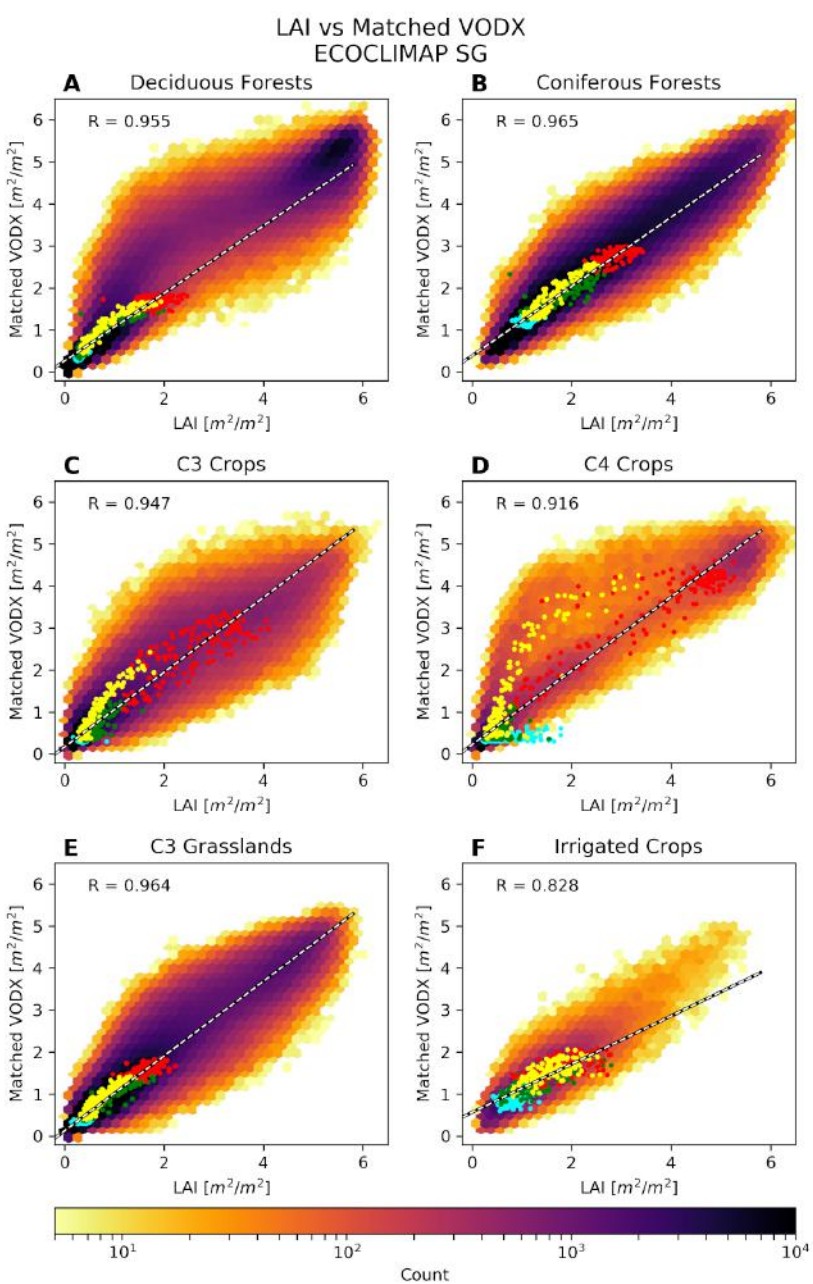

**Figure 6.** A density scatter plot detailing the relationship between LAI and Matched VODX from VODCA over six dominant vegetation types, A) Deciduous Forests, B) Coniferous Forests, C) C3 Crops, D) C4 Crops, E) C3 Grasslands, and F) Irrigated Crops. Dominant vegetation is defined as where 50% or more of a patch containing a single vegetation type. Higher concentrations of points trend towards black. Colored dots represent the spatial average each time an LAI-VOD comparison is made over the four seasons, where cyan is winter, green is spring, red is summer, and yellow is autumn. Black and white dashed lines represent the linear regression of the seasonal scores.







**Figure 7.** Graphs of monthly correlations over CONUS between LDAS-Monde OL (blue), SEKF LAI (green), SEKF VODX (red), and SEKF VODX10 (maroon) and satellite derived observations of A) LAI, B) GPP, C) ET, and D) SSM.





**Figure 8.** Graphs of probability distribution function (PDF) correlation distributions over CONUS between LDAS-Monde OL (blue), SEKF LAI (green), SEKF VODX (red), and SEKF VODX10 (maroon) and satellite derived observations of A) LAI, B) GPP, C) ET, and D) SSM. The PDFs were calculated using a Gaussian kernel density estimation of the scores. The kernel density estimation smoothing bandwidth is calculated using the default "Scott's Rule".





**Figure 9.** Probability distribution functions of the distribution of correlation differences between OL and SEKF SSM (blue), SEKF LAI (green), SEKF VODX (red), SEKF LAI SSM (cyan), and SEKF VODX SSM (orange) for USCRN at A) WG3 (5cm), B) WG_20 (20cm), C) WG6 (50cm), and D) WG8 (100cm).





**Figure 10.** Maps of Normalized Information Contribution (NIC) correlation for A-C) WG3 (5cm depth) and D-F) WG_20 (20cm) between the OL and SEKF SSM, SEKF LAI SSM, and SEKF VODX SSM. Circles represent a change greater than 3%, while triangles indicate changes less than 3%. Blue indicates correlation improvement from the assimilation with respect to the OL, while red indicates degradation.





**Figure 11.** Same as Figure 10, but for WG6 (50cm) and WG8 (100cm).






**Figure 12.** Graphs of monthly correlations over CONUS between LDAS-Monde OL (blue), SEKF SSM (cyan), SEKF LAI (green, dashed), SEKF VODX (red, dashed), SEKF LAI SSM (green, solid), and SEKF VODX SSM (red, solid) and satellite derived observations of A) LAI, B) GPP, C) ET, and D) SSM.





**Table 1.** List of experiment names and their assimilated observations analyzed

| Model | Assimilated Observations | Model Equivalents of Observations | Control Variables |
|---|---|---|---|
| ISBA-A-gs, NIT, Diffusion | LAI (CGLS), VOD (VODCA), SSM (CGLS, ESA-CCI) | LAI (for LAI and VOD) Soil Layer WG2 (1-4cm) | LAI, Soil Layers WG2-WG8 (1-100cm) |





**Table 2.** List of experiment names and their assimilated observations analyzed

| # | Experiment Name | Assimilated Observations |
|---|---|---|
| 1 | Open Loop (OL) | No Assimilation |
| 2 | SEKF LAI | CGLS LAI |
| 3 | SEKF VODX | VODCA Matched VODX |
| 4 | SEKF SSM | ESA CCI SSM |
| 5 | SEKF VODX10 | VODCA Matched VODX only when there is an associated LAI observation (every 10 days) |
| 6 | SEKF LAI SSM | Joint CGLS LAI + ESA CCI SSM |
| 7 | SEKF VODX SSM | Joint VODCA Matched VODX + ESA CCI SSM |
| 8 | SEKF VODX10 SSM | Joint VODCA Matched VODX10 + ESA CCI SSM |





**Table 3.** Average correlations scores between USCRN in situ soil moisture observations and LDAS-Monde soil moisture at 5, 20, 50, and 100cm depths. Bolded values indicate the highest score at each depth. *WG_20 is a weighted average of WG4 and WG5 in order to directly compare to 20cm observations from USCRN.

| Experiment | WG3 (5cm) (n=110) | WG_20* (20cm) (n=87) | WG6 (50cm) (n=85) | WG8 (100cm) (n=84) |
|---|---|---|---|---|
| OL | 0.75 | 0.68 | 0.59 | 0.46 |
| SEKF SSM | 0.75 | 0.69 | 0.60 | 0.46 |
| SEKF LAI | 0.75 | 0.69 | 0.60 | **0.48** |
| SEKF VODX | 0.75 | 0.69 | 0.60 | **0.48** |
| SEKF VODX10 | 0.75 | 0.69 | 0.60 | **0.48** |
| SEKF LAI SSM | 0.75 | **0.70** | **0.61** | **0.48** |
| SEKF VODX SSM | 0.75 | **0.70** | 0.60 | **0.48** |
| SEKF VODX10 SSM | 0.75 | **0.70** | **0.61** | **0.48** |





**Table 4.** Number of degraded (red), neutral (black), and improved (green) USCRN stations after assimilation using NIC R between OL and various LDAS-Monde experiments at 5, 20, 50, and 100cm depths. Stations are considered improved if the NICR is greater than 3, degraded if the score is less than 3, and neutral if it is between -3 and 3. *WG_20 is a weighted average of WG4 and WG5 in order to directly compare to 20cm observations from USCRN.

| Experiment | WG3 (5cm) (n=110) | WG_20* (20cm) (n=87) | WG6 (50cm) (n=85) | WG8 (100cm) (n=84) |
|---|---|---|---|---|
| SEKF SSM | 3/79/28 | 4/59/24 | 8/59/18 | 15/52/17 |
| SEKF LAI | 10/69/31 | 10/51/26 | 8/49/28 | 14/43/27 |
| SEKF VODX | 13/55/42 | 10/40/37 | 14/36/35 | 17/35/32 |
| SEKF VODX10 | 9/68/33 | 7/53/27 | 9/52/24 | 13/44/27 |
| SEKF LAI SSM | 7/57/46 | 6/41/40 | 12/41/32 | 13/36/35 |
| SEKF VODX SSM | 8/45/57 | 10/34/43 | 15/36/34 | 17/35/32 |
| SEKF VODX10 SSM | 7/56/47 | 7/44/36 | 13/38/34 | 17/37/30 |





$$NIC_R = \frac{R_{Analysis} - R_{Model}}{1 - R_{Model}} \times 100 \tag{1}$$