# Peer review of "Assimilation of passive microwave vegetation optical depth in LDAS-Monde: a case study over the continental US"

_Biogeosciences, 2021_

## Author Comment (AC1)

**Reviewer #1:**

*The paper illustrates how satellite-based vegetation data assimilation, and joint vegetation and soil moisture assimilation has an impact on evapotranspiration, gross primary productivity and soil moisture over CONUS. The paper is of general interest to the scientific audience, but needs some clarifications and could also benefit from a careful language check.*

**We thank the referee #1 for her/his positive comments about our work and for her/his detailed review that has helped us to improve the quality of our manuscript. Responses to comments and subsequent changes are detailed below.**

Major points:

**Comment 1.1:**

*The intense massaging of the VOD data begs the question which signal is ultimately helping the assimilation system in this paper. It would be good to make this clearer in the manuscript. The rescaling of VOD to LAI needs to be spelled out more precisely. Is a relationship found pixel per pixel and per season, or was the relationship based on the CONUS clouds per vegetation class for all months or only for the growing season or per season presented in Fig 5, or still something else? Unless I missed it, there is also no mentioning of how the discrepancy in spatial resolution (and the spatial-temporal collocation in general) in VOD and LAI data is handled to obtain this linear relationship. After rescaling and applying a 90-day rolling average, the short-term variability is probably gone. Yet, the latter might be very important to catch the start of the growing season. In the end, it sounds like the only 'original' signal that can improve the assimilation system is of an interannual (and perhaps seasonal) nature. If that is correct, it needs to be explicitly mentioned in the paper. Finally, the 90-day rolling average means that the benefit of a filter is defeated: why not simply directly use a smoother and limit the observation preprocessing?*

**Response 1.1:**

**The authors agree that the original description of the VOD re-scaling is insufficient. To remedy this, the following text has replaced a phrase in section 2.3.3:**

**"Before linear re-scaling, the LAI and VOD observations are first scaled and matched to the same 0.25° x 0.25° grid. A linear monthly re-scaling was then performed using a 3-month moving window period to best match the two datasets over seasonal timescales. Over an entire year, this re-scaling is represented by 12 monthly equations each taking into account the climatologies of the months preceding and succeeding it, and it is applied on a per pixel basis. Each monthly equation is the same from one year to another. Each equation results from a first-order linear regression. In addition to this CDF-matching, a 30-day rolling average is applied after the re-scaling to smooth the resulting LAI proxy, and allow for better performance of the assimilated data. VOD is sensitive to short term changes in vegetation water content such as rainwater interception (Saleh et al. 2006). This day-to-day variability does not reflect changes in LAI."**

Regarding the application of the re-scaling, the manuscript now describes that the LAI and VOD observations are first matched to the same spatial grid at 0.25 degree resolution. As this re-scaling is performed per monthly, with a 3-month period, the a and b parameters in the linear re-scaling are calculated with all the available observations of LAI and VOD. Additionally, several paragraphs in this section 2.3.3 have been reordered in order to first describe VODCA, and then discuss the processing of the VOD data.

It must be noticed that the original manuscript mistakenly described a 90-day rolling average applied after the re-scaling in order to smooth the results, allowing for better assimilation performance. However, in fact it was a 30-day rolling average that was applied, which is now accurately reflected in the manuscript. This 30-day rolling average does still remove some short term variability as you have noted, but it is also significantly less than the mistakenly written 90-day rolling average.

See also Response 2.1 (to Reviewer 2) regarding the difficulties in physically simulating VOD and justifying the use of a statistical re-scaling approach.

**Comment 1.2**:

Related to the above, the choice to evaluate the results only in terms of Pearson *correlation needs to be explained. Is it not more common to evaluate at least soil moisture in terms of anomaly correlations? And how about including an evaluation in terms of unbiased RMSD, or at least mention if the story remains the same for other metrics?*

**Response 1.2:**

A sentence is now added in section 2.5 explaining the choice of the correlation as a statistical metric, as well as saying that the Root Mean Squared Deviation (RMSD) was also taken, with very similar results, but is not shown:

"The correlation is chosen as it is a simple, yet effective, measure of proximity to reference datasets. The average correlation as well as distribution of correlations can allow the quick assessment of improvement or degradation, and are consistent with previous studies of LSMs and LSVs. Root mean squared deviation (RMSD) was also calculated for the comparisons to reference observations, showing the same results as with correlation, and are thus not shown."

In this work we preferred using R than anomaly correlation because the previous study from Albergel et al. (2018) has shown that NIC for anomaly correlation is too optimistic (i.e. less stations presented negative NIC values for anomaly correlation NIC).

**Comment 1.3:**

*The monthly correlations (fig 7, 12) are not clear: (i) are all time steps included, i.e. both forecasts and analysis time steps, and at which temporal resolution, (ii) are these values spatial correlations between simulations and reference data, temporally averaged per month, or (iij)*

*are these values multi-year temporal correlations computed at each location and then spatially averaged? It would be nice to also (1) compute confidence intervals for these monthly correlations; (2) show the number of pixels involved per month (the high correlations for SSM in the winter month might be applicable to far less pixels, if QC screening was applied).*

**Response 1.3:**

**These correlations are produced by combining all points in the domain into a single, long, time series, where the correlation is then computed against the observations processed in an identical way. This provides only one correlation score over the domain for each period. However, the significance of the score is strengthened due to the very large sample length (15 years are considered over a large domain containing more than 24000 land grid cells at a spatial resolution of 0.25 x 0.25 degrees). This description has also been added to the manuscript text in section 2.5.**

**"When calculating correlation to satellite-based observations of LAI, ET, GPP, and SSM, the correlations are produced by combining all points in the domain into a single, long, time series, where the correlation is then computed against the observations processed in an identical way. This provides only one correlation score over the domain for each period. However, the significance of the score is strengthened due to the far larger sample length (15 years are considered over a large domain containing more than 24000 land grid cells at a spatial resolution of 0.25 x 0.25 degrees)."**

**Comment 1.4:**

*Please read the manuscript thoroughly another time. There is some imprecise language and there are plenty of grammar issues. Random examples are listed here (the paper is full of issues; far too many to start noting):*

- *L. 5-L.8: This capability -> this positive impact (implicit flow of thoughts)*

- *L. 5 difference between model simulations and forecasts (drop forecasts?)*

- *L. 8: due to the low temporal…, [which is] at best [,] every ten days, and can suffer*

- *L.13: far more .. than.. product\*s\* (or \*an\* optical product…)*

- *L.110: for nature tiles. What is "nature"? There is a hint on line 135, that the model converts urban to bare rock – and rocks are nature?*

- *L. 119: NIT option not explained.*

- *L. 176: LAI that has been of direct estimations… (rephrase?).*

- *L. 182: large -> long wavelengths*

- *L. 192: VODX and VODC are not known to everyone, introduce*

- *L. 290: mention spatial resolution of simulations?*

- *L. 387: , and all the model, and even all the observations… (rephrase)*

- *L. 553: first sentence is poorly constructed, rephrase.*

- *Throughout: use the same number of significant numbers in the text and figures (we have everything from 0.8, to 0.66 to 0.795 for R-values)*

**Response 1.4:**

**Changes have been made to correct the noted problems. See below for each individual correction.**

- *L. 5-L.8: This capability -> this positive impact (implicit flow of thoughts)*

**"capability" has been changed to "positive impact"**

- *L. 5 difference between model simulations and forecasts (drop forecasts?)*

**"and forecasts" has been removed from this sentence**

- *L. 8: due to the low temporal…, [which is] at best [,] every ten days, and can suffer*

**L.8 now reads: "However, this positive impact does not reach its full potential due to the low temporal availability of optical-based LAI observations, which is at best, every ten days, and can suffer from months of no data over regions and seasons with heavy cloud cover such as winter or monsoon conditions."**

- *L.13: far more .. than.. product\*s\* (or \*an\* optical product…)*

**L.13 now reads: "The Vegetation Optical Depth Climate Archive (VODCA) dataset provides near-daily observations of vegetation conditions, far more frequently than optical based products such as LAI."**

- *L.110: for nature tiles. What is "nature"? There is a hint on line 135, that the model converts urban to bare rock – and rocks are nature?*

**Nature tiles are all non-urban surfaces. L. 110 now begins: "For nature (i.e. non-urban) tiles as determined by land use databases…"**

- *L. 119: NIT option not explained.*

**The NIT option allows for the simulation of non-woody above ground biomass, both leaf and structural, as well as transition the LAI variable from being prescribed to diagnostic based on the leaf biomass. This is described in L. 119-120. NIT itself is not an acronym.**

- *L. 176: LAI that has been of direct estimations… (rephrase?).*

**L. 176 now reads: "Previous implementations of LDAS-Monde have directly assimilated LAI products from optical observations."**

- *L. 182: large -> long wavelengths*

**"large" has been replaced with "long"**

- *L. 192: VODX and VODC are not known to everyone, introduce*

**Section 2.3.3 now includes the following line: "VOD is separated into wavelength bands based on the radiation wavelengths from which they are derived. This study examines C-band (3.75 to 7.50 cm) and X-band (2.50 to 3.75 cm) VOD, while also discussing L-band (15 - 30 cm) VOD."**

▪ *L. 290: mention spatial resolution of simulations?*

**Section 2.5 now begins with the following line: "The experiments performed and reported in this study occur over the Contiguous United States (CONUS) from 2003 to 2018 at 0.25° x 0.25° spatial resolution."**

▪ *L. 387: , and all the model, and even all the observations… (rephrase)*

**L. 387 now reads: "On average, the month of May sees some of the fastest vegetation change of the year for CONUS."**

▪ *L. 553: first sentence is poorly constructed, rephrase.*

**L. 553, section 5, now reads: "This study finds a generally positive relationship between observations of LAI and VODX."**

▪ *Throughout: use the same number of significant numbers in the text and figures (we have everything from 0.8, to 0.66 to 0.795 for R-values)*

**All reported scores in the text and tables are now given with two significant figures.**

**Comment 1.5:**

*L. 58: Assimilation here assumes a dynamic vegetation model, which is not present in all LSMs. In the broad sense, LAI assimilation could also refer to an updating of input LAI parameters.*

**Response 1.5:**

**L. 58 now specifies that this is applicable with LSMs capable of dynamically simulating vegetation.**

**"LAI, for example, can be constrained indirectly in LSMs capable of dynamically simulating vegetation, through the assimilation of LSVs such as brightness temperature (Vreugdenhil et al., 2016; Sawada et al., 2020) and radar backscatter (Lievens et al., 2017; Shamambo et al., 2019)"**

**Comment 1.6:**

*L. 107: is the same 20% error applied to actual LAI observations and VOD observations that are rescaled to LAI? Or did you 'rescale' the observation error somehow? Figure 4 implicitly shows that the observation error (relative to the model LAI) will be different for both. It would be nice to check the error between the model LAI and the observed LAI and the LAI-rescaled VOD and at least correct the observation error accordingly to interpret the results.*

**Response 1.6:**

In reality, in Barbu et al. (2011) and Fairbairn et al. (2017) both background and observation error standard deviations were represented in the same way for LAI. A 20% observation error was applied to LAI values larger than 2 m2m-2. A constant error value of 0.4 m2m-2 was used for LAI values below 2 m2m-2. In this study, we follow the approach proposed by Albergel et al. (2017): background LAI model errors are prescribed as in Barbu et al. (2011) and Fairbairn et al. (2017) but LAI observation errors are fixed as 20% of observed LAI values. The LAI proxy derived from the re-scaled VOD is also assimilated with these prescribed observation error standard deviation of 20%. This has been stated at the end of section 2.1 in the updated manuscript:
"In Barbu et al. (2011) and Fairbairn et al. (2017) both background and observation LAI error standard deviations were represented in the same way. A 20% error was applied to LAI values larger than 2 m2m-2. A constant error value of 0.4 m2m-2 was used for LAI values below 2 m2m-2. In this study, we follow the approach proposed by Albergel et al. (2017): background LAI model errors are prescribed as in Barbu et al. (2011) and Fairbairn et al. (2017) but LAI observation errors are fixed as 20% of observed LAI values. The LAI proxy derived from the re-scaled VOD observations is also assimilated with these prescribed observation error standard deviation of 20%. Further work would be required to assess to what extent this value of 20% is applicable to the re-scaled VOD."

**Comment 1.7:**

*L. 156: hard to believe that the CCI product provides \*daily\* data from 1978 onwards. If so, then some interpolation must have happened, and it would not be recommended to assimilate interpolated data.*

**Response 1.7:**

We agree. Not all time periods from 1978 are covered by daily data. "daily" was deleted. While the temporal sampling of the merged CCI SM product is 1 day (section 7.2.1 in [https://esa-soilmoisture-cci.org/sites/default/files/documents/public/CCI%20SM%20v06.1%20documentation/ESA_CCI_SM_RD_D2.1_v2_ATBD_v06.1_issue_1.1.pdf](https://esa-soilmoisture-cci.org/sites/default/files/documents/public/CCI%20SM%20v06.1%20documentation/ESA_CCI_SM_RD_D2.1_v2_ATBD_v06.1_issue_1.1.pdf)), not all days have associated observations. Figure 6 in the CCI Product Validation and Intercomparison Report ([https://esa-soilmoisture-cci.org/sites/default/files/documents/public/CCI%20SM%20v06.1%20documentation/ESA_CCI_SM_D4.1_v2_PVIR_v6.1_issue_1.0.pdf](https://esa-soilmoisture-cci.org/sites/default/files/documents/public/CCI%20SM%20v06.1%20documentation/ESA_CCI_SM_D4.1_v2_PVIR_v6.1_issue_1.0.pdf)) shows the fractional number of valid soil moisture observations per month over the globe.

**Comment 1.8:**

*ALEXI and FLUXCOM both use MODIS LAI-related data at some point. Would you expect even more consistency with these 'reference products' when assimilating MODIS LAI? What is the issue about data access for FLUXCOM? (this is really for the FLUXCOM developers - I want to raise awareness for open data access).*

**Response 1.8:**

It is likely that using MODIS LAI would allow for more consistency with the ALEXI and FLUXCOM products. Assimilating CGLS instead of MODIS LAI allows a more independent evaluation from ALEXI and FLUXCOM. However, a comparison assimilating MODIS LAI as well as using it for the linear re-scaling could be an interesting future pursuit. Regarding the data access for FLUXCOM, when conducting the analysis, the data was not accessible, and only the data that had been already downloaded was used. However, it looks like the FTP is now running without any access issues.

**Comment 1.9:**

*The text jumps from Fig. 9 to Fig 12; Fig 10-11 are only discussed later. Re-order the figures; perhaps the latter figures can even be removed and be presented in a table (~ Table 4).*

**Response 1.9:**

Figures 10 and 11 were moved to the Supplement as Tables 3 and 4 have the relevant information regarding the average correlations and the number of improvements and degradations.

**Comment 1.10:**

*The impact on SSM is negligible in this paper and not all in line with other studies. Is the system designed to minimally update SM, i.e. to avoid harm? How general is the conclusion that vegetation DA has a greater impact? Is it just for the ISBA model or would you expect it to be general for all LSMs?*

**Response 1.10:**

To the best of our knowledge, many similar studies assimilating SSM using state of the art land surface models (e.g. Martens et al. 2007, de Rosnay et al. 2013) obtained the same kind of results. The following paragraph has been added to the Discussion (now in section 4.3):

"The small impact of assimilating SSM can be explained by the fact that we use a state-of-the-art land surface model able to represent diffusion processes into the soil. In dry conditions, the simulated SSM is decoupled from soil moisture of deeper soil layers. As a result, assimilating SSM has a limited impact on the model state variables (Parrens et al. 2014). On the other hand, directly assimilating LAI impacts deep soil layers and a more efficient analysis of root-zone soil moisture can be done than assimilating SSM alone (see also Fig. 4 in Albergel et al. 2017)."

**References:**

De Rosnay, P.; Drusch, M.; Vasiljevic, D.; Balsamo, G.; Albergel, C.; Isaksen, L. A simplified Extended Kalman Filter for the global operational soil moisture analysis at ECMWF. Q. J. R. Meteorol. Soc. 2013, 139, 1199–1213.

Martens, B., Miralles, D. G., Lievens, H., van der Schalie, R., de Jeu, R. A. M., Fernández-Prieto, D., Beck, H. E., Dorigo, W. A., and Verhoest, N. E. C.: GLEAM v3: satellite-based land evaporation and root-zone soil moisture, Geosci. Model Dev., 10, 1903–1925, https://doi.org/10.5194/gmd-10-1903-2017, 2017.

Parrens, M., J.-F. Mahfouf, A. Barbu, and J.-C. Calvet: Assimilation of surface soil moisture into a multilayer soil model: design and evaluation at local scale, Hydrol. Earth Syst. Sci., 18, 673-689, https://doi.org/10.5194/hess-18-673-2014, 2014.

**Comment 1.11:**

*L. 505: why is there a discussion about L-band VOD if no L-band VOD is used in this paper? Similarly, why is section 4.3 in this paper?*

**Response 1.11:**

**In this paper, we mention L-band VOD as there may be links between the research conducted with L-band and other microwave frequencies in the context of vegetation data assimilation. See also Response 2.1 to Reviewer 2.**

**Section 4.3 was originally in the article to identify concrete next steps taken for this research. However, as the specifics of future drought monitoring studies are not necessary for this article, we have removed this section, and instead added the following addition to the Discussion section 4.3, formerly 4.2:**

**"By improving initial conditions of the LDAS, next steps also include testing drought forecasting by combining these known improvements through more frequent and joint assimilation of observations with LDAS-Monde's forecast capacity. The analysis of drought forecast accuracy and potential warning time could prove useful for agricultural managers and stakeholders.**

---

## Author Response (AR1)

**Editor**

*Thank you for your thorough responses to the referee reports. I partly agree with the comments of the referees but I think they can be addressed by 1) clarifying the differences between X-VOD and L-VOD already in the introduction (provide the motivation why X-VOD is used as proxy for LAI), and 2) by improving the description of the VOD-LAI scaling. Please include all your proposed changes in the revised manuscript.*

**Response:**

**Many thanks for these suggestions.**

**In response to your first comment, the following paragraph was included in the Introduction (L. 79-85 of the revised manuscript):**

*"Linking VOD to other vegetation indices has been the source of many previous studies. Saatchi et al. (2011) demonstrates that L-band satellite radar estimations of above ground biomass (AGB) are strongly impacted by forest structure, and Mialon et al. (2020) shows poor correlations between L-band VOD and estimated AGB over heavily forested areas of the Northern hemisphere. Additionally, Rodríguez-Fernández et al. (2018) and Scholze et al. (2019) found that L-band VOD conveys large amounts of information relative to AGB, primarily related to wood biomass in forested areas. Teubner et al. (2021) found that while X-band VOD correlates well with in situ FLUXNET observations of GPP, L-band VOD is poorly correlated to GPP over either low of high vegetation types."*

**In response to your second comment, the paragraph presenting Figure 2 in Section 2.3.3 was completed and moved at the end of the section. The Python code segment responsible for re-scaling of VOD values is now given in the supplement to this article.**

**See L. 229-248 of the revised manuscript:**

*"Figure 2 shows the time series response of CGLS LAI (green, solid), VODCA VODX (red, dashed), and VODCA VODC (blue, dotted) near Lincoln, Nebraska from 2003-2018. This pixel is composed primarily of C3 and C4 crops. LAI observations have a far more predictable and seasonal pattern. X-band VOD also is a stronger signal compared to C-band. The peaks are relatively close in timing in this case, but can also be offset due to the difference in peak vegetation water content. While this figure demonstrates that there is a correlation between LAI and VOD, it also shows that one cannot be substituted for the other. As was done in Kumar et al. (2020), VOD observations are seasonally linearly re-scaled to match observed LAI over the same period, in this case from the CGLS LAI dataset. Before linear re-scaling, the LAI and VOD observations are first scaled and matched to the same 0.25 x 0.25 grid. A linear monthly re-scaling was then performed using a 3-month moving window period to best match the two datasets over seasonal timescales. Over an entire year, this re-scaling is represented by 12 monthly equations each taking into account the climatologies of the months preceding and succeeding it, and it is applied on a per pixel basis. Each monthly equation is the same from one year to another. Each equation results from a first-order linear regression. In addition to this CDF-matching, a 30-day rolling average is applied after the re-scaling to smooth the resulting LAI proxy, and allow for better performance of the assimilated data. VOD is sensitive*

*to short term changes in vegetation water content such as rainwater interception (Saleh et al., 2006). This day-to-day variability does not reflect changes in LAI. For the sake of clarity regarding the re-scaling methodology, the Python code segment responsible for re-scaling of VOD values is given in the supplement to this article. Re-scaling is required because the ISBA LSM cannot simulate VOD directly, and thus we cannot assimilate VOD data directly into the model. As shown in the Figure 2 time series, as well as what was demonstrated in Albergel et al. (2018a), LAI and VOD observations are correlated and this relationship enables us to match the VOD to LAI observations and use the resulting product to assimilate in place of LAI in the model."*

**Reviewer 1**

**Comment 1.1:**

*The intense massaging of the VOD data begs the question which signal is ultimately helping the assimilation system in this paper. It would be good to make this clearer in the manuscript. The rescaling of VOD to LAI needs to be spelled out more precisely. Is a relationship found pixel per pixel and per season, or was the relationship based on the CONUS clouds per vegetation class for all months or only for the growing season or per season presented in Fig 5, or still something else? Unless I missed it, there is also no mentioning of how the discrepancy in spatial resolution (and the spatial-temporal collocation in general) in VOD and LAI data is handled to obtain this linear relationship. After rescaling and applying a 90-day rolling average, the short-term variability is probably gone. Yet, the latter might be very important to catch the start of the growing season. In the end, it sounds like the only 'original' signal that can improve the assimilation system is of an interannual (and perhaps seasonal) nature. If that is correct, it needs to be explicitly mentioned in the paper. Finally, the 90-day rolling average means that the benefit of a filter is defeated: why not simply directly use a smoother and limit the observation preprocessing?*

**Response 1.1:**

**The authors agree that the original description of the VOD re-scaling is insufficient. To remedy this, the paragraph presenting Figure 2 in Section 2.3.3 was completed and moved at the end of the section. The Python code segment responsible for re-scaling of VOD values is now given in the supplement to this article.**

**See L. 229-248 of the revised manuscript:**

*"Figure 2 shows the time series response of CGLS LAI (green, solid), VODCA VODX (red, dashed), and VODCA VODC (blue, dotted) near Lincoln, Nebraska from 2003-2018. This pixel is composed primarily of C3 and C4 crops. LAI observations have a far more predictable and seasonal pattern. X-band VOD also is a stronger signal compared to C-band. The peaks are relatively close in timing in this case, but can also be offset due to the difference in peak vegetation water content. While this figure demonstrates that there is a correlation between LAI and VOD, it also shows that one cannot be substituted for the other. As was done in Kumar et al. (2020), VOD observations are seasonally linearly re-scaled to match observed LAI over the same period, in this case from the CGLS LAI dataset. Before linear re-scaling, the LAI and VOD observations are first scaled and matched to the same 0.25 x 0.25 grid. A linear monthly re-scaling was then performed using a 3-month moving window period to best match the two datasets over seasonal timescales. Over an entire year, this re-scaling is represented by 12 monthly equations each taking into account the climatologies of the months preceding and succeeding it, and it is applied on a per pixel basis. Each monthly equation is the same from one year to another. Each equation results from a first-order linear regression. In addition to this CDF-matching, a 30-day rolling average is applied after the re-scaling to smooth the resulting LAI proxy, and allow for better performance of the assimilated data. VOD is sensitive to short term changes in vegetation water content such as rainwater interception (Saleh et al.,*

*2006). This day-to-day variability does not reflect changes in LAI. For the sake of clarity regarding the re-scaling methodology, the Python code segment responsible for re-scaling of VOD values is given in the supplement to this article. Re-scaling is required because the ISBA LSM cannot simulate VOD directly, and thus we cannot assimilate VOD data directly into the model. As shown in the Figure 2 time series, as well as what was demonstrated in Albergel et al. (2018a), LAI and VOD observations are correlated and this relationship enables us to match the VOD to LAI observations and use the resulting product to assimilate in place of LAI in the model."*

**Comment 1.2**:

Related to the above, the choice to evaluate the results only in terms of Pearson *correlation needs to be explained. Is it not more common to evaluate at least soil moisture in terms of anomaly correlations? And how about including an evaluation in terms of unbiased RMSD, or at least mention if the story remains the same for other metrics?*

**Response 1.2:**

**The second paragraph of section 2.5 was reworded. See L. 320-326 of the revised manuscript:**

*"The primary statistical score used in this study is the Pearson's correlation coefficient (R). The correlation is chosen as it is a simple, yet effective, measure of proximity to reference datasets. The average correlation as well as distribution of correlations can allow the quick assessment of improvement or degradation, and are consistent with previous studies of LSMs and LSVs. Root mean squared deviation (RMSD) was also calculated for the comparisons to reference observations, showing the same results as with correlation, and are thus not shown. In addition to the correlation, a normalized information contribution (NIC) is calculated for R as shown in Equations 1. This NICR, following Kumar et al. (2009), is normalized and thus allows for inter-comparison while accounting for differences between variables and regions."*

**In this work we preferred using R than anomaly correlation because the previous study from Albergel et al. (2018) has shown that NIC for anomaly correlation is too optimistic (i.e. less stations presented negative NIC values for anomaly correlation NIC).**

**Comment 1.3:**

*The monthly correlations (fig 7, 12) are not clear: (i) are all time steps included, i.e. both forecasts and analysis time steps, and at which temporal resolution, (ii) are these values spatial correlations between simulations and reference data, temporally averaged per month, or (iij) are these values multi-year temporal correlations computed at each location and then spatially averaged? It would be nice to also (1) compute confidence intervals for these monthly correlations; (2) show the number of pixels involved per month (the high correlations for SSM in the winter month might be applicable to far less pixels, if QC screening was applied).*

**Response 1.3:**

These correlations are produced by combining all points in the domain into a single, long, time series, where the correlation is then computed against the observations processed in an identical way. This provides only one correlation score over the domain for each period. However, the significance of the score is strengthened due to the very large sample length (15 years are considered over a large domain containing more than 24000 land grid cells at a spatial resolution of 0.25 x 0.25 degrees). This description has also been added to the manuscript text in section 2.5. See L. 327-331 of the revised manuscript:

*"When calculating correlation to satellite-based observations of LAI, ET, GPP, and SSM, the correlations are produced by combining all points in the domain into a single, long, time series, where the correlation is then computed against the observations processed in an identical way. This provides only one correlation score over the domain for each period. However, 330 the significance of the score is strengthened due to the far larger sample length (15 years are considered over a large domain containing more than 20000 pixels)."*

**Comment 1.4:**

*Please read the manuscript thoroughly another time. There is some imprecise language and there are plenty of grammar issues. Random examples are listed here (the paper is full of issues; far too many to start noting).*

**Response 1.4:**

**Changes have been made to correct the noted problems. Remaining problems would be solved after the final BG copy editing phase. See below for each individual correction in the revised manuscript.**

- ▪ *L. 5-L. 8: This capability -> this positive impact (implicit flow of thoughts) and L. 5 difference between model simulations and forecasts (drop forecasts?)*

**See L. 3-7 of the revised manuscript:**

*"It jointly assimilates satellite-derived observations of leaf area index (LAI) and surface soil moisture (SSM) into the Interactions between Soil Biosphere and Atmosphere (ISBA) land surface model (LSM), increasing the 5 accuracy of the model simulations of the LSVs. The assimilation of vegetation variables directly impacts RZSM through seven control variables consisting in soil moisture of seven soil layers from the soil surface to 1 m depth. This positive impact is particularly useful in dry conditions, where SSM and RZSM are decoupled to a large extent."*

- • *L. 8: due to the low temporal…, [which is] at best [,] every ten days, and can suffer*

**L. 8 now reads:**

*"However, this positive impact does not reach its full potential due to the low temporal availability of optical-based LAI observations, which is at best, every ten days, and can suffer*

*from months of no data over regions and seasons with heavy cloud cover such as winter or monsoon conditions."*

- *L.13: far more .. than.. product\*s\* (or \*an\* optical product…)*

**L.13 now reads:**

*"The Vegetation Optical Depth Climate Archive (VODCA) dataset provides near-daily observations of vegetation conditions, far more frequently than optical based products such as LAI."*

- *L.110: for nature tiles. What is "nature"? There is a hint on line 135, that the model converts urban to bare rock – and rocks are nature?*

**Nature tiles are all non-urban surfaces. Section 2.2.1 now begins with (L. 123):**

*"For nature (i.e. non-urban) tiles as determined by land use databases, the ISBA LSM simulates heat, carbon, water, and other surface fluxes."*

- *L. 119: NIT option not explained.*

**The NIT option allows for the simulation of non-woody above ground biomass, both leaf and structural, as well as transition the LAI variable from being prescribed to diagnostic based on the leaf biomass. This is described in L. 131-134 of the revised manuscript. NIT itself is not an acronym.**

- *L. 176: LAI that has been of direct estimations… (rephrase?).*

**This sentence now reads (L. 188 of revised manuscript):**

*"Previous implementations of LDAS-Monde have directly assimilated LAI products from optical observations."*

- *L. 182: large -> long wavelengths*

**"large" has been replaced with "long" (see L. 194 of revised manuscript).**

- *L. 192: VODX and VODC are not known to everyone, introduce*

**Section 2.3.3 now includes the following line (L. 200-202 of revised manuscript):**

*"VOD is separated into wavelength bands based on the radiation wavelengths from which they are derived. This study examines C-band (3.75 to 7.50 cm) and X-band (2.50 to 3.75 cm) VOD, while also discussing L-band (15 - 30 cm) VOD."*

- *L. 290: mention spatial resolution of simulations?*

**Section 2.5 now begins with the following line (L. 308 of revised manuscript):**

*"The experiments performed and reported in this study occur over the Contiguous United States (CONUS) from 2003 to 2018 at 0.25° x 0.25° spatial resolution."*

- *L. 387: , and all the model, and even all the observations… (rephrase)*

**L. 415 of revised manuscript now reads:**

*"On average, the month of May sees some of the fastest vegetation change of the year for CONUS."*

▪ *L. 553: first sentence is poorly constructed, rephrase.*

**L. 580 of revised manuscript, section 5, now reads:**

*"This study finds a generally positive relationship between observations of LAI and VODX."*

▪ *Throughout: use the same number of significant numbers in the text and figures (we have everything from 0.8, to 0.66 to 0.795 for R-values)*

**All reported scores in the text and tables are now given with two significant figures.**

**Comment 1.5:**

*L. 58: Assimilation here assumes a dynamic vegetation model, which is not present in all LSMs. In the broad sense, LAI assimilation could also refer to an updating of input LAI parameters.*

**Response 1.5:**

**L. 58 now specifies that this is applicable with LSMs capable of dynamically simulating vegetation.**

*"LAI, for example, can be constrained indirectly in LSMs capable of dynamically simulating vegetation, through the assimilation of LSVs such as brightness temperature (Vreugdenhil et al., 2016; Sawada et al., 2020) and radar backscatter 60 (Lievens et al., 2017; Shamambo et al., 2019)."*

**Comment 1.6:**

*L. 107: is the same 20% error applied to actual LAI observations and VOD observations that are rescaled to LAI? Or did you 'rescale' the observation error somehow? Figure 4 implicitly shows that the observation error (relative to the model LAI) will be different for both. It would be nice to check the error between the model LAI and the observed LAI and the LAI-rescaled VOD and at least correct the observation error accordingly to interpret the results.*

**Response 1.6:**

**In reality, in Barbu et al. (2011) and Fairbairn et al. (2017) both background and observation error standard deviations were represented in the same way for LAI. A 20% observation error was applied to LAI values larger than 2 m2m-2. A constant error value of 0.4 m2m-2 was used for LAI values below 2 m2m-2. In this study, we follow the approach proposed by Albergel et al. (2017): background LAI model errors are prescribed as in Barbu et al. (2011) and Fairbairn et al. (2017) but LAI observation errors are fixed as 20% of observed LAI values. The LAI proxy derived from the re-scaled VOD is also assimilated with these prescribed observation error standard deviation of 20%. This has been stated at the end of section 2.1, L. 118-121 of revised manuscript:**

*"Additionally, when assimilating re-scaled VOD observations, the same 20% error is applied as with LAI assimilation. The first assumption was to apply to the rescaled VOD the same error as for LAI, that had been proposed by Barbu et al. (2011) and subsequently applied by Fairbairn et al. (2017). Further work would be required to assess to what extent this value is applicable to the re-scaled VOD."*

**Comment 1.7:**

*L. 156: hard to believe that the CCI product provides \*daily\* data from 1978 onwards. If so, then some interpolation must have happened, and it would not be recommended to assimilate interpolated data.*

**Response 1.7:**

**We agree. Not all time periods from 1978 are covered by daily data. "daily" was deleted (L. 168 of revised manuscript).**

**While the temporal sampling of the merged CCI SM product is 1 day (section 7.2.1 in [https://esa-soilmoisture-cci.org/sites/default/files/documents/public/CCI%20SM%20v06.1%20documentation/ESA_CCI_SM_RD_D2.1_v2_ATBD_v06.1_issue_1.1.pdf](https://esa-soilmoisture-cci.org/sites/default/files/documents/public/CCI%20SM%20v06.1%20documentation/ESA_CCI_SM_RD_D2.1_v2_ATBD_v06.1_issue_1.1.pdf)), not all days have associated observations. Figure 6 in the CCI Product Validation and Intercomparison Report ([https://esa-soilmoisture-cci.org/sites/default/files/documents/public/CCI%20SM%20v06.1%20documentation/ESA_CCI_SM_D4.1_v2_PVIR_v6.1_issue_1.0.pdf](https://esa-soilmoisture-cci.org/sites/default/files/documents/public/CCI%20SM%20v06.1%20documentation/ESA_CCI_SM_D4.1_v2_PVIR_v6.1_issue_1.0.pdf)) shows the fractional number of valid soil moisture observations per month over the globe.**

**Comment 1.8:**

*ALEXI and FLUXCOM both use MODIS LAI-related data at some point. Would you expect even more consistency with these 'reference products' when assimilating MODIS LAI? What is the issue about data access for FLUXCOM? (this is really for the FLUXCOM developers - I want to raise awareness for open data access).*

**Response 1.8:**

**It is likely that using MODIS LAI would allow for more consistency with the ALEXI and FLUXCOM products. Assimilating CGLS instead of MODIS LAI allows a more independent evaluation from ALEXI and FLUXCOM. However, a comparison assimilating MODIS LAI as well as using it for the linear re-scaling could be an interesting future pursuit. Regarding the data access for FLUXCOM, when conducting the analysis, the data was not accessible, and only the data that had been already downloaded was used. However, it looks like the FTP is now running without any access issues.**

**Comment 1.9:**

*The text jumps from Fig. 9 to Fig 12; Fig 10-11 are only discussed later. Re-order the figures; perhaps the latter figures can even be removed and be presented in a table (~ Table 4).*

**Response 1.9:**

**Figures 10 and 11 were deleted.**

**Comment 1.10:**

*The impact on SSM is negligible in this paper and not all in line with other studies. Is the system designed to minimally update SM, i.e. to avoid harm? How general is the conclusion that vegetation DA has a greater impact? Is it just for the ISBA model or would you expect it to be general for all LSMs?*

**Response 1.10:**

**To the best of our knowledge, many similar studies assimilating SSM using state of the art land surface models (e.g. Martens et al. 2007, de Rosnay et al. 2013) obtained the same kind of results. The following paragraph has been added to the Discussion (now in section 4.3, L. 565-569):**

*"The small impact of assimilating SSM can be explained by the fact that we use a state-of-the-art land surface model able to represent diffusion processes into the soil. In dry conditions, the simulated SSM is decoupled from soil moisture of deeper soil layers. As a result, assimilating SSM has a limited impact on the model state variables (Parrens et al., 2014). On the other hand, directly assimilating LAI impacts deep soil layers and a more efficient analysis of root-zone soil moisture can be done than assimilating SSM alone (see also Fig. 4 in Albergel et al. (2017))."*

**References:**

**De Rosnay, P.; Drusch, M.; Vasiljevic, D.; Balsamo, G.; Albergel, C.; Isaksen, L. A simplified Extended Kalman Filter for the global operational soil moisture analysis at ECMWF. Q. J. R. Meteorol. Soc. 2013, 139, 1199–1213.**

**Martens, B., Miralles, D. G., Lievens, H., van der Schalie, R., de Jeu, R. A. M., Fernández-Prieto, D., Beck, H. E., Dorigo, W. A., and Verhoest, N. E. C.: GLEAM v3: satellite-based land evaporation and root-zone soil moisture, Geosci. Model Dev., 10, 1903–1925, https://doi.org/10.5194/gmd-10-1903-2017, 2017.**

**Parrens, M., J.-F. Mahfouf, A. Barbu, and J.-C. Calvet: Assimilation of surface soil moisture into a multilayer soil model: design and evaluation at local scale, Hydrol. Earth Syst. Sci., 18, 673-689, https://doi.org/10.5194/hess-18-673-2014, 2014.**

**Comment 1.11:**

*L. 505 (section 4.1): why is there a discussion about L-band VOD if no L-band VOD is used in this paper? Similarly, why is section 4.4 in this paper?*

**Response 1.11:**

**In this paper, we mention L-band VOD as there may be links between the research conducted with L-band and other microwave frequencies in the context of vegetation data assimilation. See also Response 2.1 to Reviewer 2. Title of Section 4.1 was changed and this section now begins with (L. 522-535 of revised manuscript):**

*"In the comparison of VOD and LAI before seasonal linear re-scaling, it is immediately apparent that vegetation type plays a large role in their relationship. These values seen and described in the results seem to indicate that heavily forested regions have only weak correlations between VOD and LAI observations. In this research, the improvements to GPP from the assimilation of X-band VOD can be explained by a better sensitivity of X-band VOD to the leaf biomass. While direct assimilation of VOD may be possible in some data assimilation systems (such as L-band VOD in CCDAS, as performed by Scholze et al. (2019)), this is not possible in LDAS-Monde, as the NIT version of ISBA simulates neither wood biomass nor specific leaf area (SLA), both necessary for simulating VOD. Additionally, the objective of VOD data assimilation in CCDAS is to constrain certain model parameters, while the objective of assimilating re-scaled X-band VOD in LDAS-Monde is to sequentially assimilate observations in order to constrain the day-to-day trajectory of the ISBA state variables, without changing model parameter values. Although ISBA is an uncalibrated model, it performs as well as other state-of-the-art models in inter-comparison experiments (e.g. Fig. B2 in Friedlingstein et al. (2020)), even without assimilation. Moreover, studies have shown that VOD may be sensitive to rainwater interception by leaves (e.g. Saleh et al. (2006)). The ISBA model is able to simulate interception, but there is no simple way to simulate the physical interception effect on VOD. It is for this reason that a statistical re-scaling of VOD towards an LAI proxy was pursued."*

**Section 4.4 was originally in the article to identify concrete next steps taken for this research. However, as the specifics of future drought monitoring studies are not necessary for this article, we have removed this section, and instead added the following addition to the Discussion section 4.3 (L. 576-578 of revised manuscript):**

*"By improving initial conditions of the LDAS, next steps also include testing drought forecasting by combining these known improvements through more frequent and joint assimilation of observations with LDAS-Monde's forecast capacity. The analysis of drought forecast accuracy and potential warning time could prove useful for agricultural managers and stakeholders."*

**Reviewer 2**

**Comment 2.1:**

*After reading this work, I admit that the data assimilation algorithm and the experiments conducted with LDAS-Monde is reasonable, but the only thing I am not convinced is the replace of LAI with VOD. This paper made the assumption mainly based on Kumar et al. (2019) with showed VOD can be seen linear with LAI. But we need to keep in mind that Kumar et al. (2019) also pointed out that VOD is different from LAI. The authors also showed in Fig.2. From a modeler's perspective, I feel this is too bold to do so and use this data for assimilation. Because this looks more like a forced matching of VOD to LAI. Some other papers (e.g. Rodríguez-Fernández et al., 2018) have pointed out that VOD contained both information about LAI and biomass, and the assimilation of VOD together with soil moisture has been successfully conducted in the Carbon Cycle Data Assimilation System (CCDAS) by Scholze et al. (2019). So I think the simulation of VOD by LSM is already possible. Therefore I do not agree that the re-scaling of VOD to LAI is due to the lack of model representation on VOD.*

**Response 2.1:**

**Your primary concern is regarding the seasonal linear re-scaling technique to match VOD to LAI, and the subsequent assimilation of the re-scaled VOD. In this research, the seasonal linear re-scaling is a statistical generalization, but it is also a method that has been nearly identically performed in Kumar et al. (2020). This article advances the same methodology by applying it to LDAS-Monde. A novelty with respect to the work of Kumar is that this assimilation directly impacts the model root-zone soil moisture layers (1-100cm), which is a unique capability of LDAS-Monde so far. The authors do fully acknowledge that VOD is not LAI. This is why a complex seasonal rescaling had to be performed to obtain a proxy of LAI from VOD. We realize that writing "linear rescaling" or "linearly rescaled" can be misleading. This has been corrected in the revised version of the paper, noting that it is a seasonal linear re-scaling. It is also true that VOD observations may convey other information such as biomass as demonstrated in Fig. 8 of Rodríguez-Fernández et al. (2018), and by Scholze et al. (2019). However, the latter studies used L-band VOD (from SMOS), while in this study X-band VOD is used. The L-band allows a much better penetration of the microwave signal through vegetation than at X-band. As a result, the latter is mainly sensitive to the leaf biomass while SMOS VOD data are mostly related to the wood biomass in forested areas. Teubner et al. (2021) showed that while X-band VOD correlates well with in situ FLUXNET observations of GPP, L-band VOD is poorly correlated to GPP over either low of high vegetation types. The better GPP-VOD correlation at X-band could be explained by a better sensitivity of X-band VOD to the leaf biomass. In the analysis of the X-band VOD vs. LAI relationship, we have found that overall, there is a link between the two variables, as also shown in previous literature. This is an argument for seasonally rescaling X-band VOD. Additionally, while it may be possible to directly assimilate L-band VOD in CCDAS as performed by Scholze et al. (2019), this is not possible in LDAS-Monde, as the NIT version of ISBA now used in LDAS-Monde does not simulates the wood biomass capable of simulating VOD, nor changes in**

specific leaf area (SLA), that would be needed to simulate VOD. Moreover, studies have shown that VOD may be sensitive to rainwater interception by leaves (e.g. Saleh et al. 2006). The ISBA model is able to simulate interception but as far as we know, there is no simple way to simulate the physical interception effect on VOD. It is for this reason that a statistical re-scaling of VOD towards an LAI proxy was pursued. As described in the discussion section, these results will be used to pave the way towards more efficient assimilation of level 1 observations using machine learning techniques.

The following paragraph was added to the Introduction (L. 79-85 of revised manuscript):

*"Linking VOD to other vegetation indices has been the source of many previous studies. Saatchi et al. (2011) demonstrates that L-band satellite radar estimations of above ground biomass (AGB) are strongly impacted by forest structure, and Mialon et al. (2020) shows poor correlations between L-band VOD and estimated AGB over heavily forested areas of the Northern hemisphere. Additionally, Rodríguez-Fernández et al. (2018) and Scholze et al. (2019) found that L-band VOD conveys large amounts of information relative to AGB, primarily related to wood biomass in forested areas. Teubner et al. (2021) found that while X-band VOD correlates well with in situ FLUXNET observations of GPP, L-band VOD is poorly correlated to GPP over either low of high vegetation types."*

The Python code segment responsible for re-scaling of VOD values is now given in the supplement to this article.

The following paragraph was added to Section 4.1 (L. 524-525 of revised manuscript):

*"In this research, the improvements to GPP from the assimilation of X-band VOD can be explained by a better sensitivity of X-band VOD to the leaf biomass."*

We have included the following references:

Kumar, S. V., Holmes, T. R., Bindlish, R., de Jeu, R., and Peters-Lidard, C.: Assimilation of vegetation optical depth retrievals from passive microwave radiometry, Hydrol. Earth Syst. Sci., 24, 3431–3450, https://doi.org/10.5194/hess-24-3431-2020, 2020.

Rodríguez-Fernández,N. J., Mialon, A., Mermoz, S., Bouvet, A., Richaume, P., Al Bitar, A., et al.: An evaluation of SMOS L-band vegetation optical depth (L-VOD) data sets:High sensitivity of L-VOD to above-ground biomass in Africa, Biogeosciences, 15, 4627–4645. https://doi.org/10.5194/bg-15-4627-2018, 2018,

Saleh, K., Wigneron, J.-P., de Rosnay, P., Calvet, J.-C., Kerr, Y., Waldteufel, P., Escorihuela, M.J.: Impact of rain interception by vegetation and mulch on the L-band emission of natural grass, Remote Sens. Env., 101, 127-139, https://doi.org/10.1016/j.rse.2005.12.004, 2006.

Scholze, M., Kaminski, T., Knorr, W., Voßbeck, M., Wu, M., Ferrazzoli, P., et al.: Mean European carbon sink over 2010–2015 estimated by simultaneous assimilation of atmospheric $CO_2$, soil moisture, and vegetation optical depth, Geophysical Research Letters, 46, https://doi.org/10.1029/2019GL085725, 2019,

Teubner, I. E., Forkel, M., Wild, B., Mösinger, L., and Dorigo, W.: Impact of temperature and water availability on microwave-derived gross primary production, Biogeosciences, 18, 3285–3308, https://doi.org/10.5194/bg-18-3285-2021, 2021.

**Comment 2.2:** *Therefore, I do not think this paper can be published in its current shape, unless the authors solve the problems in simulating VOD by the LSM. Before doing that, I do not think the detailed comments on the context is helpful for the authors, even I made some from my side.*

**Response 2.2:**

**In order to avoid misunderstandings, we have emphasize that assimilation in LDAS-Monde differs from the assimilation in CCDAS. While the objective of CCDAS is to constrain model parameters values, LDAS-Monde consists of sequentially assimilating observations in order to constrain the day-to-day trajectory of the ISBA state variables, without changing model parameter values. Although being an uncalibrated model, ISBA performs as well as other state-of-the-art models in intercomparison experiments (e.g. Fig. B2 in Friedlingstein et al. 2020), even without assimilation. In order to improve the paper, we propose rewording parts of the discussion section to better describe the shortcomings of the re-scaling methodology, and how it is leading to more efficient assimilation of microwave level 1 observations in future studies.**

**The following has been added in section 4.1 (L. 526-535 of revised manuscript):**

*"While direct assimilation of VOD may be possible in some data assimilation systems (such as L-band VOD in CCDAS, as performed by Scholze et al. (2019)), this is not possible in LDAS-Monde, as the NIT version of ISBA simulates neither wood biomass nor specific leaf area (SLA), both necessary for simulating VOD. Additionally, the objective of VOD data assimilation in CCDAS is to constrain certain model parameters, while the objective of assimilating re-scaled X-band VOD in LDAS-Monde is to sequentially assimilate observations in order to constrain the day-to-day trajectory of the ISBA state variables, without changing model parameter values. Although ISBA is an uncalibrated model, it performs as well as other state-of-the-art models in inter-comparison experiments (e.g. Fig. B2 in Friedlingstein et al. (2020)), even without assimilation. Moreover, studies have shown that VOD may be sensitive to rainwater interception by leaves (e.g. Saleh et al. (2006)). The ISBA model is able to simulate interception, but there is no simple way to simulate the physical interception effect on VOD. It is for this reason that a statistical re-scaling of VOD towards an LAI proxy was pursued."*

**We have include the following reference:**

Friedlingstein, P., O'Sullivan, M., Jones, M. W., Andrew, R. M., Hauck, J., Olsen, A., Peters, G. P., Peters, W., Pongratz, J., Sitch, S., Le Quéré, C., Canadell, J. G., Ciais, P., Jackson, R. B., Alin, S., Aragão, L. E. O. C., Arneth, A., Arora, V., Bates, N. R., Becker, M., Benoit-Cattin, A., Bittig, H. C., Bopp, L., Bultan, S., Chandra, N., Chevallier, F., Chini, L. P., Evans, W., Florentie, L., Forster, P. M., Gasser, T., Gehlen, M., Gilfillan, D., Gkritzalis, T., Gregor, L., Gruber, N., Harris, I., Hartung, K., Haverd, V., Houghton, R. A., Ilyina, T., Jain, A. K., Joetzjer, E., Kadono, K., Kato, E., Kitidis, V., Korsbakken, J. I.,

Landschützer, P., Lefèvre, N., Lenton, A., Lienert, S., Liu, Z., Lombardozzi, D., Marland, G., Metzl, N., Munro, D. R., Nabel, J. E. M. S., Nakaoka, S.-I., Niwa, Y., O'Brien, K., Ono, T., Palmer, P. I., Pierrot, D., Poulter, B., Resplandy, L., Robertson, E., Rödenbeck, C., Schwinger, J., Séférian, R., Skjelvan, I., Smith, A. J. P., Sutton, A. J., Tanhua, T., Tans, P. P., Tian, H., Tilbrook, B., van der Werf, G., Vuichard, N., Walker, A. P., Wanninkhof, R., Watson, A. J., Willis, D., Wiltshire, A. J., Yuan, W., Yue, X., and Zaehle, S.: Global Carbon Budget 2020, Earth Syst. Sci. Data, 12, 3269–3340, https://doi.org/10.5194/essd-12-3269-2020, 2020.

---

## Author Response (AR2)

**Associate Editor**

*Please implement the requested changes of referee #1 in your manuscript.*

**Response:**

**We have implemented the changes requested by referee #1. We have also revised the English, especially in the Introduction, and made editorial changes in the Abstract.**

**Reviewer 1**

**Comment 1.1:**

*Overall, the authors have addressed all comments, but some points could still be improved. The explanation on the rescaling is appreciated, but answers like 'remaining problems would be solved after the final BG copy editing phase' and the fact that the Fluxcom website is running, but the authors did not update their results, are somewhat disappointing. Therefore, I strongly encourage the authors to update the results for the extended GPP data; a revision is an ideal chance for it to make the paper more wholesome.*

**Response 1.1:**

**Many thanks for your suggestions aiming at further improving the manuscript.**

**In our response we wrote 'Changes have been made to correct the noted problems. Remaining problems would be solved after the final BG copy editing phase.' and we acknowledge that this is not clear. It should read: "We did our best to correct typos and the English. The final copy editing phase will help finalize this process." We have revised the English, especially in the Introduction, and made editorial changes in the Abstract.**

**In our response we wrote 'Regarding the data access for FLUXCOM, when conducting the analysis, the data was not accessible, and only the data that had been already downloaded was used. However, it looks like the FTP is now running without any access issues.' Unfortunately, the first author has changed position and we do not have resources to integrate more FLUXCOM years into the analysis right now. It must be noticed that there was a word of caution that interannual variability patterns of FLUXCOM data may not be completely realistic (Jung et al. 2020).**

**We propose to replace on L. 285 of the revised manuscript**

**"FLUXCOM GPP is available globally at 0.5° x 0.5° resolution, and from 1980 to present, however this study only uses data up to 2013 due to lack of data access."**

**by**

*"When this study was performed, the global FLUXCOM GPP data at 0.5° x 0.5° resolution were available from 1980 to 2013. It must be noticed that there was a word of caution that*

*interannual variability patterns of FLUXCOM data may not be completely realistic (Jung et al. 2020).”*

**For Figures 4 to 10, the periods of time that are considered are now indicated.**

**Reference:**

**Jung, M., Schwalm, C., Migliavacca, M., Walther, S., Camps-Valls, G., Koirala, S., Anthoni, P., Besnard, S., Bodesheim, P., Carvalhais, N., Chevallier, F., Gans, F., Goll, D. S., Haverd, V., Köhler, P., Ichii, K., Jain, A. K., Liu, J., Lombardozzi, D., Nabel, J. E. M. S., Nelson, J. A., O'Sullivan, M., Pallandt, M., Papale, D., Peters, W., Pongratz, J., Rödenbeck, C., Sitch, S., Tramontana, G., Walker, A., Weber, U., and Reichstein, M.: Scaling carbon fluxes from eddy covariance sites to globe: synthesis and evaluation of the FLUXCOM approach, Biogeosciences, 17, 1343–1365, https://doi.org/10.5194/bg-17-1343-2020, 2020.**

**Comment 1.2**:

*It would also be good to have a more nuanced view on the impact of soil moisture data assimilation. There are many studies that have shown benefits of assimilating SMOS, SMAP or ASCAT SSM retrievals (or even brightness temperature) to improve soil moisture in state-of-the-art land surface models (often over the US, Canada, Australia). It is not trivial why this study shows comparatively so little impact and it would be great for future researchers to frame this result a little more, i.e. to clarify an apparent contradiction. Perhaps it is just that there are less good reference data in Europe, or less variability in climate, or is the vertical coupling strength of the ISBA model just less than other models?*

**Response 1.2:**

[revised manuscript text omitted]